# PHYWORLDBENCH: A COMPREHENSIVE EVALUATION OF PHYSICAL REALISM IN TEXT-TO-VIDEO MODELS

**Jing Gu[†], Xian Liu[‡], Yu Zeng[‡], Ashwin Nagarajan[†], Fangrui Zhu[§], Daniel Hong[†], Yue Fan[†]**
**Qianqi Yan[†], Kaiwen Zhou[†], Ming–Yu Liu[‡,\*], Xin Eric Wang[†,¶,\*]**

[†] University of California, Santa Cruz  [‡] NVIDIA Research
[§] Northeastern University; [¶] University of California, Santa Barbara; [\*] Co-advisors

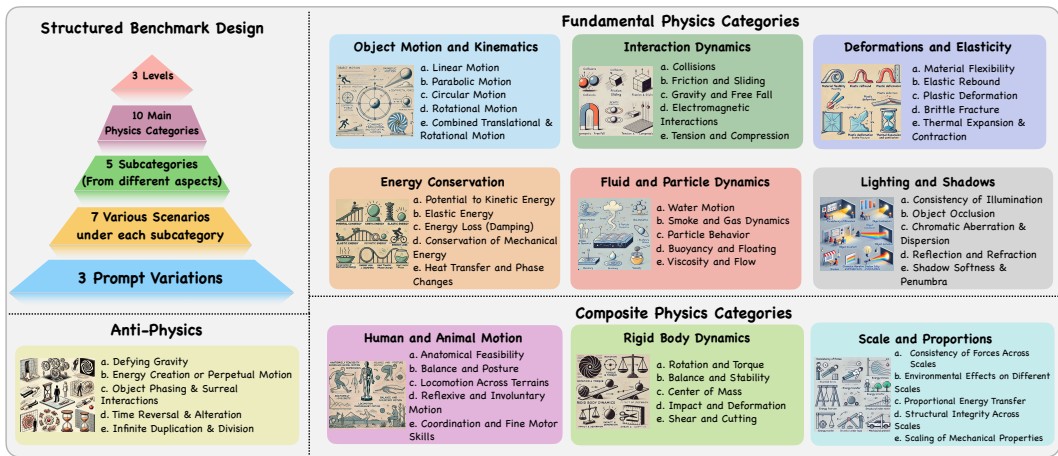

Figure 1: **Overview of PhyWorldBench.** The benchmark follows a structured design, starting with 10 main physics categories, derived from physics literature and expert consultations. Each category is divided into 5 subcategories, capturing different aspects. Under each subcategory, 7 scenarios are created, with 3 prompt variations per scenario to provide varying levels of detail and complexity. The figure presents the benchmark structure, showcasing the 10 main categories and their corresponding 5 subcategories.

## ABSTRACT

Video generation models have achieved remarkable progress in creating high-quality, photorealistic content. However, their ability to accurately simulate physical phenomena remains a critical and unresolved challenge. This paper presents `PhyWorldBench`, a comprehensive benchmark designed to evaluate video generation models based on their adherence to the laws of physics. The benchmark covers multiple levels of physical phenomena, ranging from fundamental principles like object motion and energy conservation to more complex scenarios involving rigid body interactions and human or animal motion. Additionally, we introduce a novel "Anti-Physics" category, where prompts intentionally violate real-world physics, enabling the assessment of whether models can follow such instructions while maintaining logical consistency. Besides large-scale human evaluation, we also design a simple yet effective method that could utilize current MLLM to evaluate the physics realism in a zero-shot fashion. We evaluate 12 state-of-the-art text-to-video generation models, including five open-source and five proprietary models, with a detailed comparison and analysis. we identify pivotal challenges models face in adhering to real-world physics. Through systematic testing of their outputs across 1,050 curated prompts—spanning fundamental, composite, and anti-physics scenarios—we identify pivotal challenges these models face in adhering to real-world physics. We then rigorously examine their performance on diverse phys-

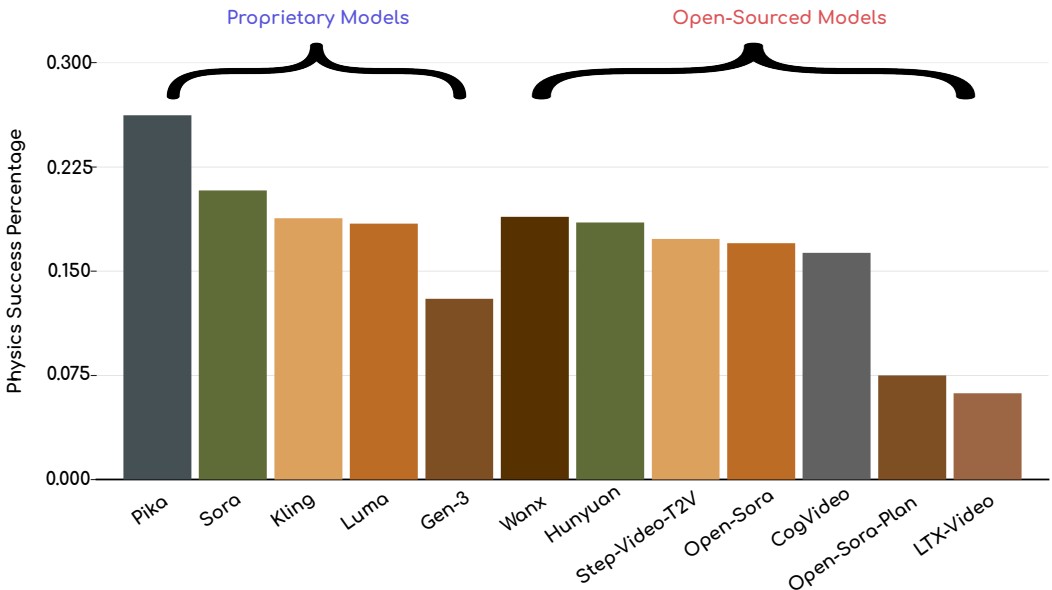

Figure 2: **Success rates of video generation models on `PhyWorldBench`.** Among open-source models, Wanx demonstrated the highest performance, while Pika achieved the best results among proprietary models with a success rate of 0.262. Despite these advancements, substantial progress remains necessary to refine the capability of these models to accurately simulate the intricate dynamics of the real world.

ical phenomena with varying prompt types, deriving targeted recommendations for crafting prompts that enhance fidelity to physical principles.

## 1 INTRODUCTION

The field of video generation has made remarkable progress, with models producing visually compelling and often photorealistic outputs. These advances have enabled transformative applications across industries such as entertainment, education, and scientific visualization. However, despite their visual fidelity, do video generation models truly understand the laws of physics in the real world? To answer this question, we introduce `PhyWorldBench`, a rigorous benchmark designed to evaluate how well video generation models can simulate **real-world physics**. As illustrated in Figure 1, `PhyWorldBench` systematically tests models across multiple levels of physical phenomena, from fundamental concepts like object motion to complex dynamics, including rigid body interactions and human/animal motion. Additionally, we propose a novel **Anti-Physics** category, where prompts deliberately violate real-world physics. On one hand, this design verifies whether models genuinely understand physical laws—rather than merely reproducing patterns from real-world training data. On the other hand, anti-physics content itself holds practical value in creative applications, where imaginative or otherwise impossible scenarios are beneficial.

We meticulously designed and annotated **1,050 prompts** and the standard set for each prompt individually to cover a broad range of physical scenarios. This substantial annotation work ensures that our benchmark is both comprehensive and precise, allowing for a more thorough assessment of video generation models' capabilities. Furthermore, we present a context-aware-prompt metric using MLLM (OpenAI Team, 2024; Gemini Team, 2024), which directly assesses if the video satisfies the physics standards or not. Such evaluation not only provided an unbiased metric but also significantly reduced the evaluation cost.

To examine the current status of video generation models and provide a detailed analysis, we selected five proprietary models—Sora-Turbo (OpenAI, 2024), Gen-3 (Runway Team, 2024), Kling 1.6 (KlingAI, 2024), Pika 2.0 (Pika Labs Team, 2024), and Luma (Luma AI Team, 2024)—along with seven open-source models: Hunyuan 720p (Kong et al., 2024), Open-Sora 2.0 (Peng et al., 2025), Open-Sora-Plan 1.3 (Lin et al., 2024), CogVideoX-1.5 (Hong et al., 2022), Step-video-T2V (Ma et al., 2025), Wanx-2.1 (WanTeam et al., 2025), and LTX-Video (HaCohen et al., 2024). We generated

12,600 videos to evaluate SOTA models and analyze their ability to simulate real-world physics. We identify challenges, difficult scenarios, and key physics categories while proposing a structured approach to improve physical realism through prompt design.

The insights obtained from this benchmark will contribute to the development of more robust and physically accurate video generation models, addressing both fundamental scientific questions and practical challenges in the field. Figure 2 presents the overall benchmark results on `PhyWorldBench`. Despite recent advancements, models continue to struggle with temporal consistency, realistic motion, and physical plausibility, emphasizing the need for further research to enhance their physics correctness for real-world simulation.

Our work provides a benchmark for video generation models with a focus on physical realism. The key contributions are:

- We propose `PhyWorldBench`, a large-scale, multi-dimensional physics benchmark for evaluating the physics ability of the video generation model.
- We conduct an extensive evaluation of twelve state-of-the-art video generation models (five proprietary and seven open-source) with 12,600 generated videos and identified key challenges in simulating real-world physics.
- We study the effect of prompt variation on the performance of the video generation model and provided prompt guidelines for generating physics-following videos.

## 2 RELATED WORK

**Benchmarks for Text-to-Video Generation.** Proprietary video generators (OpenAI, 2024; KlingAI, 2024; Pika Labs Team, 2024; Runway Team, 2024; HailuoAI, 2024; Luma AI Team, 2024) achieve striking quality and temporal coherence but remain opaque. Open-source counterparts (Peng et al., 2025; Lin et al., 2024; Hong et al., 2022; Wang et al., 2023; HaCohen et al., 2024; Kong et al., 2024; Agarwal et al., 2025) enable reproducible research and flexible benchmarking. The development of Text-to-Video (T2V) Generation models has been accelerated by benchmarks that evaluate performance across diverse aspects like quality, realism, and compositionality. VBench (Huang et al., 2024) and EvalCrafter (Liu et al., 2024) provide comprehensive frameworks for assessing video generation models, focusing on metrics such as diversity, temporal coherence, and scalability. Besides, T2V-CompBench (Sun et al., 2024) emphasizes compositionality, ensuring generated videos align with intricate textual prompts, a key challenge in real-world applications. Recent works emphasize the importance of physical plausibility in video generation. Morpheus Zhang et al. (2025a) focuses on experimental physics. VideoPhy (Bansal et al., 2024) and PhyGenBench (Meng et al., 2024) assess models' adherence to physical commonsense, while DEVIL (Liao et al., 2024) and VideoPhy-2 (Bansal et al., 2025) evaluates dynamic realism, including motion and temporal consistency. Physics-IQ (Motameda et al., 2025) further tests understanding of fundamental principles like fluid dynamics and thermodynamics, highlighting the demand for physics-aware video generation. Many physics-focused video generation models are also built (Zhang et al., 2025b; Yuan et al., 2025; Wang et al., 2025).

We introduce a novel and **substantially expanded dataset** for evaluating video generation models across a broader range of physical laws. Unlike prior benchmarks, it includes diverse, carefully curated scenarios with both real and deliberately unphysical dynamics. Models perform significantly worse on our benchmark under identical settings, revealing its greater difficulty and realism.

**Evaluation Metrics for Text-to-Video Generation.** Traditional text-to-video evaluation metrics like Fréchet Video Distance (FVD) (Unterthiner et al., 2018) and Inception Score (IS) (Salimans et al., 2016) focus on visual quality but fail to assess motion plausibility and adherence to physical laws, especially without reference videos. Recent methods address this gap: VideoScore (He et al., 2024) integrates human feedback, T2V-CompBench (Sun et al., 2024) uses VLMs for spatial and temporal evaluation, and PhyGenBench (Meng et al., 2024) explicitly tests physical commonsense. However, these methods can be computationally intensive or subjective. We introduce a Yes/No answering metric for efficient, scalable, and objective evaluation of physical commonsense in video generation. Additionally, we propose a simple yet effective strategy leveraging MLLMs like GPT-o1 to assess video quality.

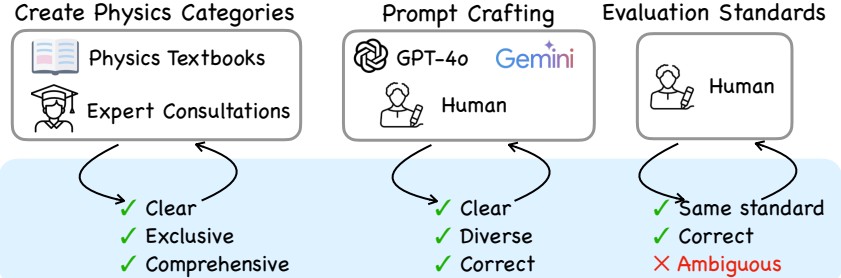

Figure 3: **Creation Process of `PhyWorldBench`.** The dataset is built through a three-stage pipeline for clarity, consistency, and completeness. First, physics categories and prompts are defined using literature and expert input. Next, GPT-4o, Gemini-1.5-Pro together with human refine prompts for diversity and accuracy. Finally, a curation phase standardizes all prompts, with human-in-the-loop reviews ensuring clarity and eliminating ambiguities.

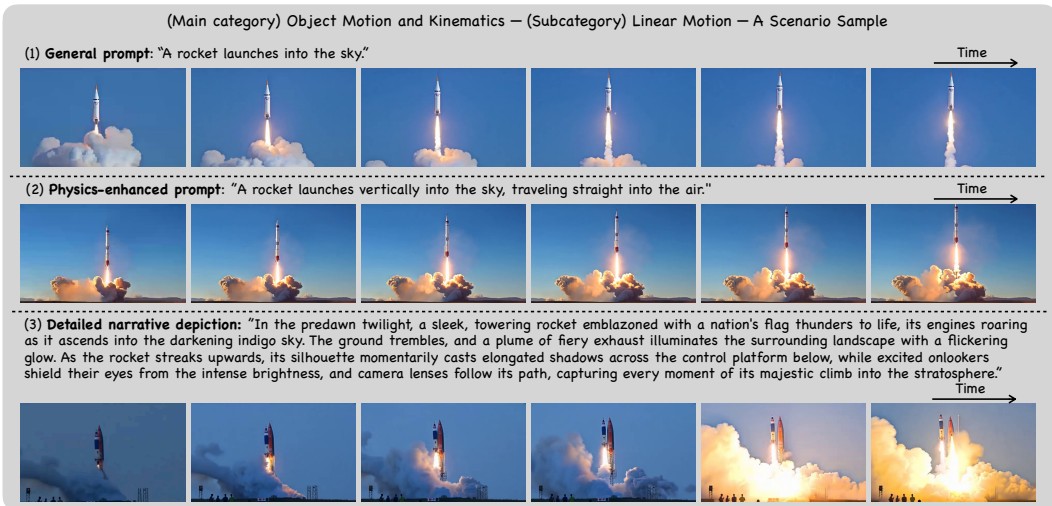

Figure 4: **Example of Three Prompt Types for a Scenario.** This figure illustrates a scenario from the subcategory *Linear Motion* under the main category *Object Motion and Kinematics*. The scenario is presented through three levels of prompts: (1) Event Prompt, providing a concise and straightforward event description; (2) Physics-Enhanced Prompt, which builds on the general prompt by incorporating physics-related phenomena while avoiding explicit physical laws; and (3) Detailed Narrative Prompt, enriching the Evnet Prompt with vivid details and contextual elements.

## 3 PHYWORLDBENCH

To evaluate the ability of text-to-video generation models to simulate physical reality, we introduce `PhyWorldBench`, a comprehensive benchmark spanning a wide range of physical principles and scenarios. Its design is rooted in fundamental physics concepts such as motion, energy conservation, and interaction dynamics, drawing from established literature, including *Fundamentals of Physics* (Halliday et al., 2013) and *Classical Mechanics* (Goldstein et al., 2002). Developed with input from experts in physics and video generation.

### 3.1 DATASET CREATION PIPELINE

As shown in Figure 3, the creation pipeline consists of three steps, with human-in-the-loop reviews integrated at each step to ensure the benchmark's accuracy and diversity.

**Step 1: Initial Physics Categories Definition.** A collaborative process between physics experts and the authors to define the *key* (main) physics categories to be included in the benchmark. Our benchmark categorizes three levels:

- **Fundamental Physics**: Covers basic laws such as object motion, energy transfer, and optics.
- **Composite Physics**: Involves real-world phenomena that emerge from multiple interacting principles, such as human motion.

Table 1: **Statistical Comparison with Existing Physics-Focused T2V Benchmarks.** Compared to prior benchmarks, `PhyWorldBench` offers a significantly larger and more diverse testbed for evaluating the physical commonsense capabilities of T2V models. It provides a broader coverage of physics categories and a greater number of prompts, ensuring a comprehensive assessment of T2V performance.

| Statistic | VideoPhy | PhyGenBench | Physics-IQ | T2VPhysBench | PhyWorldBench |
|---|---|---|---|---|---|
| **Physics Categories** | 5 | 27 | 5 | 3 | 50 |
| **Prompts** | 688 | 160 | 396 | 84 | 1050 |

- **Anti-Physics**: Scenarios intentionally designed to violate real-world physics.

The inclusion of anti-physics cases is crucial, as discrepancies in model performance between physically accurate and unphysical scenarios reveal whether a model truly understands physics or merely replicates patterns from its training data, which predominantly follows real-world laws.

We identify broad categories—such as kinematics, rigid body dynamics, fluid behavior, optics, and thermal dynamics—as depicted in the main physics categories in Figure 1. Each category is systematically divided into **five** subcategories to ensure comprehensive coverage of physics principles from multiple perspectives.

**Step 2: Prompt Creation** Each subcategory is further expanded into **7** distinct scenarios, with each scenario incorporating three prompt variations: **Event Prompt**, **Physics-enhanced Prompt**, and **Detailed Narrative Prompt**, as illustrated in Figure 4. Specifically, we define these three types of prompts as follows:

- **Event Prompt**: A concise and straightforward description of an event. These prompts are designed to assess the model's ability to generate based on minimal input, focusing on high-level scene comprehension.
- **Physics-Enhanced Prompt**: This type incorporates physics-related phenomena for Event Prompt to enrich the description naturally.
- **Detailed Prompt**: Inspired by how current T2V models generate videos, these prompts are enriched with vivid details and contextual elements, aiming to create more immersive and visually rich outputs. This level of prompt allows us to assess whether providing richer context leads to improved physical accuracy in video generation.

We first use state-of-the-art large language models (LLMs), specifically GPT-4o (OpenAI Team, 2024) and Gemini-1.5-Pro (Gemini Team, 2024) to generate 20 Event Prompts under the predefined category and subcategory. Then we ask humans to select 7 out of them based on diversity and physics representation. Then human experts meticulously review and validate the generated prompts, providing feedback to iteratively enhance their quality and relevance to the corresponding physics subcategory, ensuring a balanced and representative assessment across various aspects of physical understanding. Furthermore, an **ethical review** is conducted to ensure that all prompts are fair, unbiased, and free from potentially harmful content. Based on the Event Prompt, human annotators create a Physics-Enhanced Prompt by enriching it with physical consequences. Detailed Narrative Prompt is generated by LLM based on Event Prompt using query in Tab. 12. This rigorous verification process results in a refined set of **1,050** high-quality prompts, systematically organized into main and subcategories.

**Step 3: Standard Creation.** We adopt a Yes/No evaluation metric to assess whether a model can generate videos that accurately align with a given prompt. This metric is chosen for its simplicity and effectiveness and for allowing for an objective assessment while minimizing subjectivity in evaluation. As shown in Figure 5, we define two types of evaluation standards: **Basic Standards** and **Key Standards**. The Basic Standards specify the essential object(s) that should appear in the video and ensure that the main action or event is depicted. Meanwhile, the Key Standards describe the key physical phenomena that would occur if the event took place in the real world. The annotators are instructed to focus on the natural physical consequences of the action and to be explicit and concise in their annotations. By following these guidelines, we ensure that the evaluation standards remain clear and objective. Following (Bansal et al., 2024; Meng et al., 2024), we use **Semantic Adherence (SA)** and **Physical Commonsense (PC)** to evaluate video performance. SA checks if both "objects" and "Action/Event" align with video, while PC assesses adherence to real-world physics and it checks

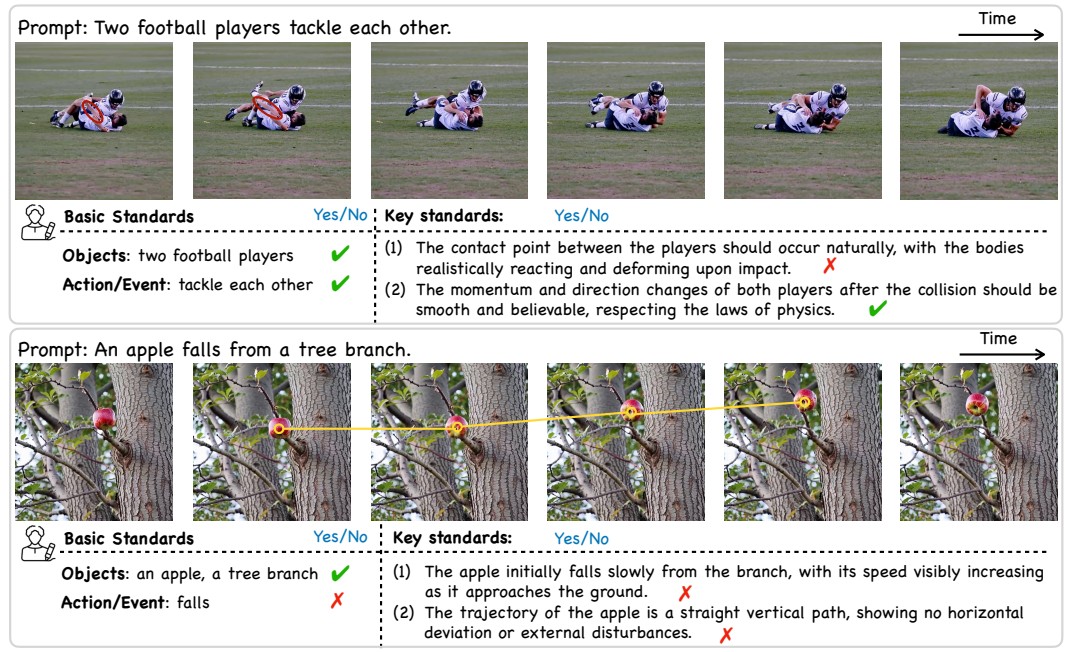

Figure 5: **Illustration of Our Evaluation Metric and Human Annotations.** We demonstrate our evaluation process for assessing the quality of generated videos based on two evaluation criteria: Basic Standards and Key Standards. For Basic Standards, we verify whether the generated video contains the correct number of objects and accurately represents the intended action or event. For Key Standards, we define specific physical phenomena as ground truth and measure if all of these phenomena the generated video satisfies. Both lead to either a score of "0" or "1" for a generated video. Red circles and yellow lines in the figure highlight instances where the generated videos fail to meet the Key Standards.

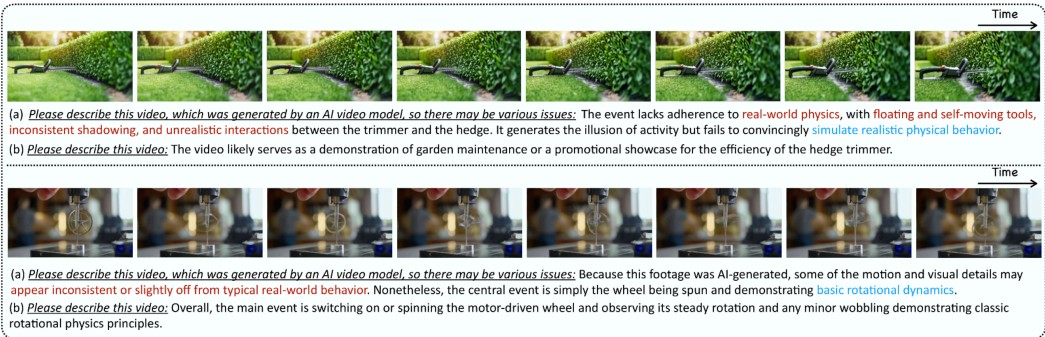

Figure 6: **Context-aware prompt improves the assessment quality for GPT-4o.** In the two examples, prompt (a) explicitly tells the MLLM that the video might contain potential issues, while prompt b does not. We found that the quality assessment is usually more accurate when the context is used. The text is irrelevant to quality is ignored for readability. This phenomenon applied to all our tested models including GPT-4o, GPT-o1, Gemini-2.0-flash, and Qwen-VL-2.0.

when all "Key Standards" are satisfied. Annotated as SA, PC $\in 0, 1$ (1 indicating proper grounding), we report the fraction of videos satisfying a success when SA $= 1$, PC $= 1$, and both jointly.

## 3.2 STATISTICS OF THE DATASET

We compare our benchmark with existing physics-focused T2V benchmarks (Bansal et al., 2024; Meng et al., 2024; Guo et al., 2025; Motameda et al., 2025), with a detailed statistical comparison presented in Tab. 1. `PhyWorldBench` offers the most extensive coverage in terms of the number of physics categories, curated prompts, human annotations, and evaluated T2V videos.

Table 2: **ROC-AUC for automatic evaluation methods and ablation on `CAP`.** Rows now list the two metrics—**SA** (*semantic adherence*) and **PC** (*physical commonsense*)—while columns compare all baselines and ablations.

| Metric | Qwen-VL-2.0 | Gemini-2.0-Flash | GPT-4o | GPT-o1 | CAP (w/o CoT) | CAP (w/o Context) | CAP |
|--------|-------------|------------------|--------|--------|----------------|--------------------|------|
| SA | 72.4 | 74.6 | 72.1 | 75.4 | 76.3 | 77.3 | **80.3** |
| PC | 59.8 | 60.9 | 60.1 | 61.6 | 73.6 | 65.6 | **75.1** |

### 3.3 MLLM AS ZERO SHOT PHYSICS EVALUATOR FOR GENERATED VIDEOS

Human evaluation, while flexible, is costly at scale. Existing metrics often lack generalizability and struggle with the diversity in `PhyWorldBench`. (See Appx. D.) MLLMs like GPT-o1 (OpenAI Team, 2024) and Gemini (Gemini Team, 2024) show promise for video evaluation but fall short in reliably assessing physical realism (Bansal et al., 2024; He et al., 2024; Meng et al., 2024). We find that these models tend to rationalize unrealistic content, but their accuracy improves when informed that the video is AI-generated, as illustrated in Figure 6. We hypothesize that this improvement stems from MLLMs being primarily trained on real-world videos, leading them to inherently justify observed phenomena unless prompted otherwise. Inspired by this, we propose a simple yet effective method, **Context-Aware Prompt** (**CAP**), which explicitly informs the MLLM that the video is generated rather than real. This approach enables the MLLM to provide a more accurate assessment of the video's physical realism. Critically, CAP does not assume videos are incorrect by default but instead uses a structured chain-of-thought prompt—first asking the model to describe the video and then to reason through any potential physics issues—and only labels a video as incorrect if it clearly violates the category-specific standards. We validate CAP neutrality by measuring false positive rates in physics correct videos and observe only a 0.001 absolute change in semantic adhesion (0.188 to 0.187) and a 0.014 change in physical commonsense (0.172 to 0.186), showing that it does not significantly increase misclassification of correct videos.

Besides, inspired by Chain-of-Thought (Wei et al., 2022), we design a two-step prompting strategy, where we first ask the MLLM to give a thorough description of the video including **Object**, **Event**, and **Observations**, then continue to prompt the MLLM for the final "Yes/No" answer for video quality. Please refer to Appx. K for prompts used in the two steps. We found this leads to consistent improvement in various MLLMs. In Tab. 5, we present the ROC AUC calculations comparing `CAP` with human results on `PhyWorldBench` with videos generated by models in Figure 2, where a random guess would achieve a score of 50. We evenly sample 8 frames. In this evaluation, `CAP` is powered by GPT-o1. `CAP` significantly outperforms other methods, particularly in Physics Commonsense, where it achieves an absolute performance boost of 13.5 compared to the standard GPT-o1. We find `CAP` also achieves great performance when equipping other proprietary or open-sourced MLLM. The bottom of Tab. 5 shows the ablation study of various components in `CAP`. "without CoT" means we directly ask the MLLM to give the answer of the question, and "without Context" means we do not inform MLLM that the video could be of low quality. Please refer Appx. D for performance of our method on other MLLM and analysis.

## 4 VIDEO GENERATION RESULTS AND ANALYSIS

We evaluate both closed-source models including Sora-Turbo (OpenAI, 2024), Gen-3 (Runway Team, 2024), Kling 1.6 (KlingAI, 2024), Pika 2.0 (Pika Labs Team, 2024), Luma (Luma AI Team, 2024), and open-source models including Hunyuan 720p (Kong et al., 2024), Open-Sora 2.0 (Peng et al., 2025), Open-Sora-Plan 1.3 (Lin et al., 2024), CogVideoX-1.5 (Hong et al., 2022), LTX-Video (HaCohen et al., 2024), Step-video-T2V (Ma et al., 2025), Wanx-2.1 (WanTeam et al., 2025). Please check Appx. E for the implementation details. We generate 1050 videos for each model. We conduct a human evaluation of generated videos using Amazon Mechanical Turk. We first introduce model performance and model-wise analysis and then break down the challenge for the video generation model to achieve real-world physics.

### 4.1 MODEL PERFORMANCE

Tab. 3 presents a comprehensive evaluation of twelve video generation models. We report the percentage of videos that satisfy both semantic adherence and physical commonsense criteria (Both).

Table 3: **We present the human evaluation results of 10 video generation models across 3 physics types.** **SA** denotes *semantic adherence*, while **PC** represents *physical commonsense*. **Both** indicates satisfaction of both criteria.

| Model | Fundamental Physics | | | Composite Physics | | | Anti-Physics | | | Overall Performance | | |
|---|---|---|---|---|---|---|---|---|---|---|---|---|
| | SA | PC | Both | SA | PC | Both | SA | PC | Both | SA | PC | Both |
| *Proprietary Models* | | | | | | | | | | | | |
| Sora-Turbo (OpenAI, 2024) | 0.438 | 0.315 | 0.246 | 0.361 | 0.215 | 0.188 | 0.136 | **0.078** | 0.039 | 0.384 | 0.261 | 0.208 |
| Gen-3 (Runway Team, 2024) | 0.320 | 0.228 | 0.161 | 0.258 | 0.142 | 0.103 | 0.108 | 0.029 | 0.020 | 0.280 | 0.183 | 0.130 |
| Kling-1.6 (KlingAI, 2024) | 0.417 | 0.299 | 0.235 | 0.312 | 0.175 | 0.139 | 0.125 | 0.073 | **0.042** | 0.357 | 0.241 | 0.188 |
| Pika 2.0 (Pika Labs Team, 2024) | **0.587** | **0.375** | **0.312** | **0.479** | **0.274** | **0.239** | **0.228** | 0.043 | 0.011 | **0.521** | **0.314** | **0.262** |
| Luma (Luma AI Team, 2024) | 0.464 | 0.259 | 0.220 | 0.328 | 0.199 | 0.160 | 0.056 | 0.000 | 0.000 | 0.385 | 0.218 | 0.184 |
| *Open-sourced Models* | | | | | | | | | | | | |
| Hunyuan 720p (Kong et al., 2024) | 0.385 | **0.278** | 0.205 | **0.342** | **0.264** | **0.199** | 0.082 | 0.021 | 0.010 | 0.344 | **0.250** | 0.185 |
| Open-Sora 2.0 (Peng et al., 2025) | **0.434** | 0.268 | 0.205 | 0.279 | 0.202 | 0.154 | 0.056 | 0.021 | 0.010 | 0.348 | 0.223 | 0.170 |
| Open-Sora-Plan 1.3 (Lin et al., 2024) | 0.175 | 0.136 | 0.096 | 0.159 | 0.057 | 0.044 | 0.069 | **0.059** | **0.039** | 0.158 | 0.104 | 0.075 |
| CogVideoX-1.5 (Hong et al., 2022) | 0.419 | 0.266 | 0.192 | 0.308 | 0.207 | 0.154 | 0.062 | 0.021 | 0.000 | **0.351** | 0.225 | 0.163 |
| Step-video-T2V (Ma et al., 2025) | 0.353 | 0.245 | 0.189 | 0.333 | 0.251 | 0.191 | **0.085** | 0.021 | 0.020 | 0.320 | 0.224 | 0.173 |
| Wanx-2.1 (WanTeam et al., 2025) | 0.389 | 0.271 | **0.220** | 0.323 | 0.235 | 0.187 | 0.082 | 0.017 | 0.011 | 0.339 | 0.235 | **0.189** |
| LTX-Video (HaCohen et al., 2024) | 0.221 | 0.102 | 0.080 | 0.177 | 0.071 | 0.042 | 0.076 | 0.011 | 0.000 | 0.194 | 0.085 | 0.062 |

The Overall Performance column provides an aggregate score for each model, averaged across 10 physics categories. Pika 2.0 achieves the best overall performance across all models. In open-sourced models, Hunyuan 720p achieves the best performance on PC, Wanx-2.1 achieves the best performance on Both, and CogVideoX-1.5 achieves the best SA performance. It is worth noting that they even achieve a better performance compared with some of the proprietary. Please check Appx. P for Leaderboard.

## 4.2 MODEL SPECIFIC ANALYSIS

Our evaluation reveals a trade-off between realism, stylization, and consistency in text-to-video models. Wanx, Step-Video-T2V, Gen-3, Hunyuan, Kling, and Sora produce the most realistic videos, while Pika and Luma add stylized lighting and colors. Open-Sora and LTX-Video struggle with realism, often generating low-fidelity or distorted outputs. Object stability varies—CogVideoX-1.5, Open-Sora, and LTX-Video frequently produce warped shapes, while Gen-3, Hunyuan, and Kling maintain more stable forms. Motion realism also differs: Kling and Gen-3 generate smooth motion, Open-Sora-Plan and Luma introduce slow-motion effects, and CogVideoX-1.5 and Open-Sora exhibit jittery, erratic movement. For a detailed analysis, see Appx. L.

## 4.3 EFFECT OF PHYSICS AND PROMPT ENHANCEMENT

Understanding real-world physics is essential for text-to-video models. Event-based prompts describe actions, while physics-enhanced prompts include natural consequences. A model's ability to generate realistic videos based on these prompts reflects its real-world understanding. As proprietary models conceal their generation processes, we cannot verify prompt refinement. Instead, we analyze prompt types using open-source models, where prompt integrity can be manually ensured. In Tab. 4, we compare model performance across different prompt types to gauge the effects of physics-aware prompting and refinement. While refinement improves clarity, object detail, and aesthetics, it does not notably enhance physics accuracy. The persistent inconsistencies suggest that these refinements primarily address surface-level quality, leaving deeper physics limitations unresolved. On the other hand, physics-enhanced prompts often yield better results, reflecting the models' strong prompt-following ability—even though they still struggle with true physical comprehension. Consequently, we propose a straightforward receipt: **explicitly integrate physical phenomena into the prompt to nudge the model toward more realistic outcomes.**

## 4.4 ANTI-PHYSICS ANALYSIS

As shown in Tab. 3, all models show a performance decline from Fundamental to Composite to Anti-Physics categories, reflecting the challenge of generating complex physical phenomena. While they capture basic physics, simulating intricate interactions and "unphysical" scenes remains difficult, highlighting a gap in physics adherence. Our analysis found that failures often stem from models interpreting prompts in a more "reasonable" way rather than attempting and failing. For example, when prompted with "A glass sits on the table, and the wine in the glass has reversed gravity," the model generates a still image, adhering to real-world physics instead of the prompt. **This suggests that models prioritize mimicking training data over true physics understanding.**

### 4.5 FAILURE TO HANDLE ERRATIC VISUAL CHANGES

Certain physical events inherently result in erratic visual changes, such as collisions, breaking, or sudden state transitions. However, current video generation models struggle to accurately depict these scenarios. In the third example of Figure 10, a glass cup falls from a table, an event that should cause it to shatter upon impact. Instead, the model rationalizes the action, generating a video where the glass remains intact. Rather than simulating realistic disruptions in structure, motion, or object state, the models tend to default to continuous, smoothed animations, avoiding abrupt transformations that are critical for representing real-world physics. We created 20 paired prompts for each test, applied them to 12 video generation models (240 videos total), and only evaluate the visual change correctness through human evaluation. We also highlight Pika, the strongest model overall. Tab. 16 shows models struggle with abrupt visual changes — only 22.1% success overall vs. 42.1% for softened prompts; Pika 30% vs. 75%.**When prompted with erratic visual changes, the model tends to simplify the task and generate the easier version.**

### 4.6 BREAKDOWN IN PHYSICS AT HIGHER COMPLEXITY

Models' performance deteriorates under complex interactions involving multiple forces or constraints. As demonstrated in Appx. R, prompts involving multiple objects introduce significantly higher requirements for semantic adherence. For instance, when with the prompt *"A feather and a stone are dropped in the air, where both fall, and the stone drops much faster"*, all models fail to present the phenomenon correctly. The current models lack a robust understanding of how multiple physical forces interact in complex environments. We similarly built 20 prompt pairs focused on interactions of multiple forces or collisions, applied them to the 12 models, and highlighting Pika as the best model as in Tab. 17. Physics further breaks down at higher complexity — 12% success overall vs. 18% for simpler cases; Pika 15% vs. 40%. This suggest that **the current video models do not really understand the physics thoroughly and are more likely to mimic the pattern in the training set.**

Table 4: **Physics-following percentage on different types of prompt.** Explicitly adding physics usually increases the physics-following ability of video generation model, while a prompt refinement process does not necessary leads to improvement.

| Model | Event Prompt | Physics-enhanced Prompt | Detailed Prompt |
|---|---|---|---|
| CogVideoX-1.5 (Hong et al., 2022) | 0.123 | **0.177** | 0.168 |
| Hunyuan 720p (Kong et al., 2024) | 0.159 | **0.198** | 0.155 |
| LTX-Video (HaCohen et al., 2024) | 0.056 | 0.065 | **0.066** |
| Open-Sora-Plan 1.3 (Lin et al., 2024) | 0.062 | **0.067** | 0.063 |
| Open-Sora 2.0 (Peng et al., 2025) | 0.167 | **0.177** | 0.173 |
| Wanx-2.1 (WanTeam et al., 2025) | 0.175 | **0.202** | 0.190 |
| Step-Video-T2V (Ma et al., 2025) | 0.158 | **0.182** | 0.179 |

### 4.7 CINEMATIC EFFECTS VS. PHYSICAL PLAUSIBILITY

A major challenge in video generation is the tendency for models to prioritize cinematic aesthetics over strict physical realism. Well-performed models like Sora, Gen-3, and Pika often introduce **exaggerated motion dynamics**, such as objects moving with heightened fluidity or unnatural acceleration, making scenes appear choreographed rather than physically grounded. For example, an object that should fall naturally under gravity might instead descend too smoothly or float unrealistically. This issue is particularly evident when momentum conservation and force dynamics are distorted for dramatic effect. The first example in Figure 11 presented an apple floating in the air, then it suddenly dropped as the camera moved; such stylized rendering, while beneficial for artistic storytelling, poses significant challenges for applications requiring strict physical fidelity.

## 5 CONTRIBUTIONS AND FUTURE WORK

We propose `PhyWorldBench`, a large-scale, thorough, and multi-dimensional benchmark for evaluating text-to-video generation models. It provides insights into which physical phenomena are hardest to simulate and what capabilities current models lack. Additionally, we propose an automated evaluator to measure physical accuracy and a straightforward receipt for designing prompt. Future work will involve continuously evaluating emerging video generation models, updating our benchmark to reflect advancements in the field, and refining our leaderboard and analysis to track progress in physical accuracy and model capabilities. LLM usage is described in Appx. A.

**Ethics statement**    The authors confirm that we have carefully reviewed and adhered to the ICLR Code of Ethics in this work.

**Reproducibility statement**    We open-sourced our codebase to ensure reproducibility. We first released all prompts, evaluation standards, and we released the full evaluation code of our benchmark. We also released all the generated videos and their corresponding prompts.

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

## A  LLM USAGE STATEMENT

The large language model is used during the manual script proofreading to detect grammatical errors and rephrase some overly long sentences. Meanwhile, as mentioned in the paper, our designed evaluator is based on LLM, the prompt draft creation process also involved LLM.

## B  BENCHMARK DETAILS

The benchmark revolves around **10** main physics categories, each divided into **5** subcategories that cover specific physics phenomena, such as object occlusion, center of mass, and circular motion. Each subcategory includes **7** distinct scenarios, with **3** variations of prompts: a general prompt, a physics-enhanced prompt, and a detailed narrative prompt. These variations are designed to provide differing levels of complexity and context for the text-to-video models. The dataset was curated through a three-stage process: first, defining key categories based on fundamental physics literature;

second, refining prompts through iterative reviews with large language models and human experts to ensure clarity and alignment with physics principles; and third, conducting detailed curation to create a final set of **1,050** high-quality prompts for a diverse set of physics phenomena. The evaluation utilizes a yes/no framework to assess model adherence to physical laws, categorized into basic standards and key standards, enabling comprehensive performance evaluation. This benchmark establishes a foundation for evaluating models' ability to generate videos that align with both physical principles and real-world scenarios.

Following Bansal et al. (2024) and Meng et al. (2024), we use **Semantic Adherence (SA)** and **Physical Commonsense (PC)** as metric to evaluate the video performance. SA measures if a caption aligns with video frames, while PC assesses if actions obey real-world physics. Annotated as $SA, PC \in \{0, 1\}$, where 1 indicates proper grounding. We report the fraction of videos satisfying $SA = 1$, $PC = 1$, and both jointly. We create SA and PC based on a thorough review of all prompts and extensive feedback from human annotators, we establish a set of evaluation criteria to systematically assess the generated videos. The evaluation is structured around two key standards, each encompassing specific rules:

- **Semantic Adherence**: These criteria evaluate fundamental aspects of video generation to ensure prompt fidelity. The assessment includes:
    1. Whether the video contains the correct number of objects specified in the prompt. (Yes/No)
    2. Whether the depicted event or action is accurately represented. (Yes/No)
- **Physical Commonsense**: This aspect evaluates the model's capability to capture and represent underlying physical phenomena within the video. The assessment measures the number of distinct physical phenomena accurately showcased in the generated video.

By integrating these two standards, which consist of three core evaluation rules, we define a comprehensive evaluation metric that ensures a balanced and thorough assessment of T2V model performance across multiple dimensions.

## C  ANNOTATION DETAILS

**Prompt Annotation** After designing the physics categories, we further prepared prompts within each category. To achieve this, we gathered undergraduate and graduate students majoring in Computer Science for an annotation session. They were instructed to provide high-quality annotations by creating both an **Event Prompt** and a corresponding **Physics-enhanced Prompt** that aligned with the primary physics category and sub-category. The **Detailed Prompt** was generated using GPT-4o with the MLLM prompt (as described in Tab. 12) and subsequently verified by human annotators. Finally, each prompt underwent a final verification by the authors.

**Standard Annotation** Undergraduate and graduate students in Computer Science were also tasked with annotating the objects and events described in the prompts. These annotations were used for SA (Semantic Adherence) evaluation. Additionally, they identified key physics phenomena that should be visually observable in the generated videos, which were used for PC (Physics Commonsense) evaluation. All annotations were reviewed and verified by the authors.

**Video Evaluation** For large-scale video evaluation, we utilized Amazon Mechanical Turk. The interface is presented in Figure 7 Annotators were asked to assess videos based on three criteria: the existence of objects, the presence of the described event, and the visibility of key physics phenomena, selecting either "Yes" or "No" for each. Each video was evaluated by three independent annotators, with the final decision determined by a majority vote. All annotators were compensated at a rate of $18 USD per hour.

## D  DETAILED ANALYSIS ON EVALUATOR

While some existing physics metrics have been proposed, none fit `PhyWorldBench`'s needs due to domain or structural mismatches (Guo et al., 2025; Zhang et al., 2025a). For example, PhyGenEval (Meng et al., 2024) assumes a fixed event order, which many of `PhyWorldBench`'s scenarios do not have, and VideoCon-Physics (Bansal et al., 2024) is trained specifically on its own

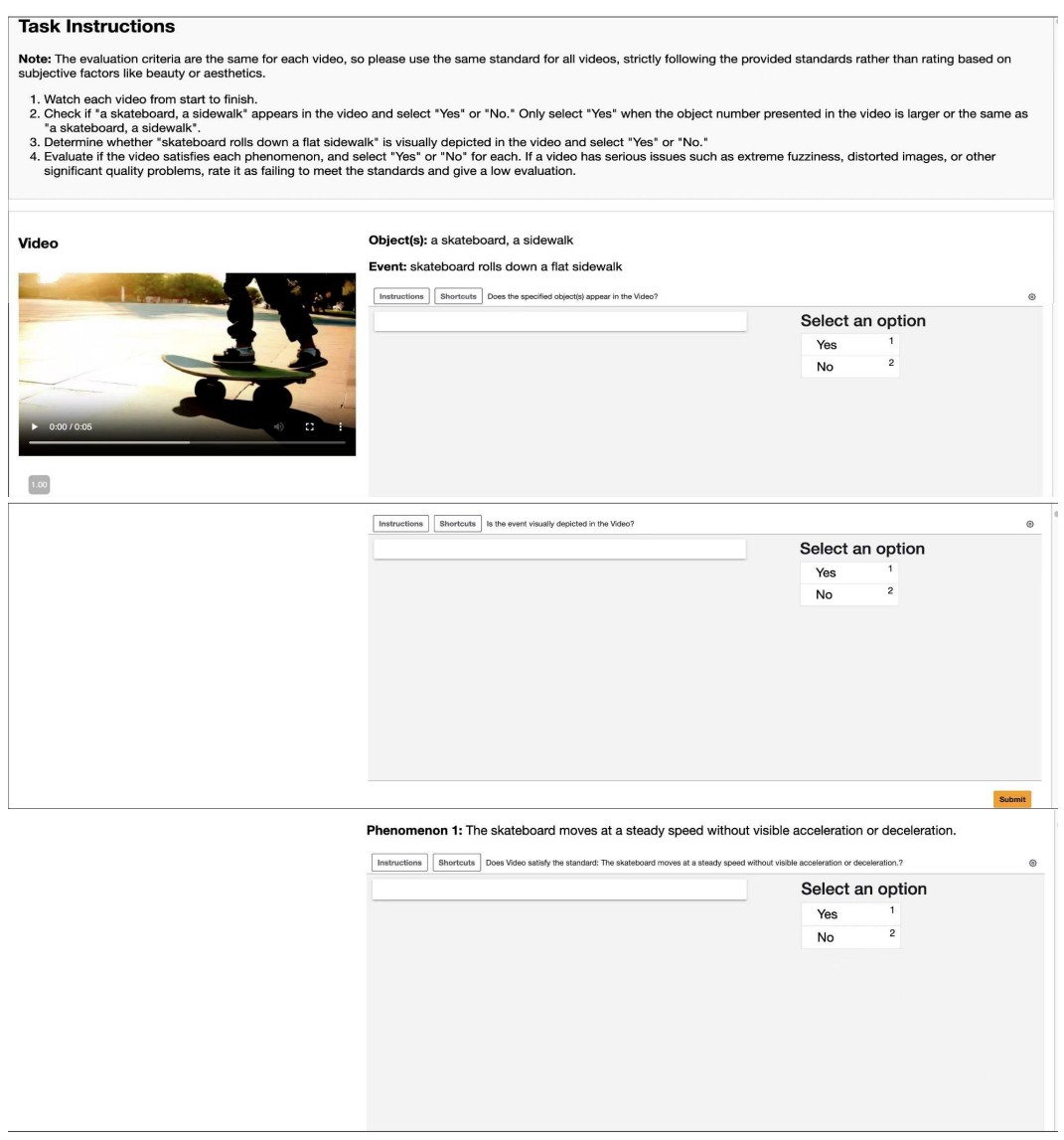

Figure 7: **Amazon Mechanical Turk interface.**

Table 5: **Precision and Recall for automatic evaluation methods and ablation on `CAP`, together with ROC-AUC from the submission.**

| Method | SA | Precision of SA | Recall of SA | PC | Precision of PC | Recall of PC |
|---|---|---|---|---|---|---|
| Qwen-VL-2.0 | 72.4 | 66.9 | 72.5 | 59.8 | 59.7 | 62.5 |
| Gemini-2.0-Flash | 74.6 | 68.0 | 73.1 | 60.9 | 61.1 | 61.8 |
| GPT-4o | 72.1 | 69.3 | 71.0 | 60.1 | 59.4 | 63.1 |
| GPT-o1 | 75.4 | 71.1 | 73.2 | 61.6 | 63.1 | 62.4 |
| CAP (without CoT) | 76.3 | 71.8 | 79.3 | 73.6 | 70.3 | 74.6 |
| CAP (without Context) | 77.3 | 72.1 | 78.8 | 65.6 | 70.1 | 71.6 |
| CAP | **80.3** | **75.8** | **83.1** | **75.1** | **74.8** | **79.3** |

VideoPhy benchmark, yielding sub-optimal performance on `PhyWorldBench`—so we exclude both to avoid unfair comparisons. Besides, VideoPhy-2 (Bansal et al., 2025) focused on action and dynamic instead of general physics, and its evaluator is also designed and finetuned on the associated benchmark annotation, therefore for the same reason, we also did not include it in this paper.

In Tab. 6, we therefore evaluate our `CAP` across multiple MLLMs, including proprietary GPT-4o, GPT-o1, Gemini-Flash-2.0, and open-source Qwen-VL-2.0. Crucially, `CAP` uses a two-step chain-of-thought prompt—first asking the model to describe the video, then to reason about any physics issues—so it does not assume videos are incorrect by default and yields only minimal false-positive increases (0.001 in Semantic Adherence, 0.014 in Physical Commonsense), confirming its neutrality. Moreover, because it operates solely on the final video and its physics-based standards, `CAP` is agnostic to the underlying generative pipeline—whether text-to-video, image-to-video, video-to-video, or hybrid.

We do observe two common failure modes: 1) Aesthetic bias, where visually compelling videos receive high scores despite only moderate physical accuracy (visual appeal and physical plausibility should ideally be assessed independently); and 2) Overconfidence, with the MLLM giving firm judgments in ambiguous or edge-case scenarios where human annotators often express uncertainty or disagreement.

To further validate our design, we also measure Precision and Recall for both Semantic Adherence (SA) and Physical Commonsense (PC), which is shown in Table 5.

Table 6: **ROC-AUC Improvement of various MLLM for with our method automatic evaluation methods.** Our method consistently achieves great performance on both open-sourced and proprietary models.

| Method | SA | PC |
|---|---|---|
| Qwen-VL-2.0 | 72.4→78.6 | 59.8→74.1 |
| Gemini-2.0-Flash | 74.6→79.9 | 60.9→74.0 |
| GPT-4o | 72.1→78.5 | 60.1→73.9 |
| GPT-o1 | 75.4→80.3 | 61.6→75.1 |

Moreover, even without any fine-tuning, our CAP evaluator achieves 80.3 Semantic Adherence and 75.1 Physical Commonsense on `PhyWorldBench`, and when this is compared to VideoCon-Physics's 82.0 SA and 73.0 PC on the VideoPhy benchmark after fine-tuning, it highlights CAP's strong out-of-domain performance. Additionally, we conducted fine-tuning on an 80/20 train/test split of our annotated videos and found that these numbers can be boosted substantially: Qwen-VL-2.0 improves from 77.5 → 82.3 SA and 74.5 → 79.1 PC, and GPT-4o from 78.5 → 84.7 SA and 73.9 → 80.7 PC.

## E MODEL APPLICATION DETAILS

**Open-source generators (code + weights).**

- **Hunyuan** (Tencent Hunyuan Community License Agreement) : based on the official 720p repo `https://github.com/Tencent/HunyuanVideo`.

Table 7: Automatic Evaluation results from `CAP` on proprietary and open-source video models.

| Model | SA | PC | Both |
|---|---|---|---|
| **Proprietary Models** | | | |
| Pika 2.0 | 0.354 | 0.250 | 0.205 |
| Luma | 0.275 | 0.177 | 0.135 |
| Kling-1.6 | 0.343 | 0.237 | 0.177 |
| Sora-Turbo | 0.357 | 0.225 | 0.176 |
| Gen-3 | 0.380 | 0.164 | 0.133 |
| **Open-sourced Models** | | | |
| Hunyuan 720p | 0.323 | 0.230 | 0.183 |
| Step-Video-T2V | 0.335 | 0.187 | 0.153 |
| Open-Sora-Plan 1.3 | 0.143 | 0.110 | 0.065 |
| Open-Sora 2.0 | 0.330 | 0.196 | 0.143 |
| CogVideoX-1.5-5B | 0.303 | 0.187 | 0.129 |
| Wanx-2.1 | 0.365 | 0.243 | 0.191 |
| LTX-Video | 0.197 | 0.075 | 0.052 |

- **LTX-Video** (RAIL-M License): Hugging Face Diffusers pipeline
  `https://huggingface.co/docs/diffusers/api/pipelines/ltx_video`.
- **CogVideo v1.5** (Apache-2.0 License): `https://github.com/THUDM/CogVideo`.
- **Open-Sora v2.0** (Apache-2.0 License): `https://github.com/hpcaitech/Open-Sora`.
- **Wanx v2.1** (Apache-2.0 License): `https://github.com/Wan-Video/Wan2.1`.
- **Step-Video-T2V** (MIT License): `https://github.com/stepfun-ai/Step-Video-T2V`.
- **Open-Sora-Plan v1.3** (MIT License): `https://github.com/PKU-YuanGroup/Open-Sora-Plan`.

**Commercial / proprietary generators.**

- **Sora-Turbo**, **Pika 2.0**, **Gen-3**, **Kling 1.6**: accessed via the vendors' public web interfaces under their standard Terms of Service (no OSS license).
- **Luma**: accessed through the official API.

All third-party code or data are *used* under the above licenses/ToS but are **not redistributed** in our release. Our own benchmark assets are released under the MIT License.

## F   `CAP` RESULTS

Here we further include the automatic evaluation results of `CAP` on the video generation models on Tab. 7.

## G   INSIGHTS INTO PERFORMANCE BETWEEN MODELS

Our analysis reveals that both model size and sampling budget significantly impact physics performance. Larger models consistently outperform smaller ones on foundational physics tasks, as shown in Table 8. For instance, the fact that Cosmos 14B outperforms Cosmos 7B across all SA/PC metrics confirming that scaling yields clear gains in semantic adherence and physical commonsense.

We also observe that generation time correlates with physics accuracy: models with longer sampling schedules (e.g., Hunyuan at 40 min/video or Wanx-2.1 at 1 hr/video) achieve stronger performance, and simply increasing Open-Sora's sampling steps from 50 to 200 yields a significant boost. These

Table 8: **Influence of model scale on physics performance.** SA denotes *semantic adherence*, PC denotes *physical commonsense*, and Both indicates satisfaction of both criteria. To isolate the effect of size, we compare variants of the same base architecture.

| Model | Fundamental Physics | | | Composite Physics | | | Anti-Physics | | | Overall Performance | | |
|---|---|---|---|---|---|---|---|---|---|---|---|---|
| | SA | PC | Both | SA | PC | Both | SA | PC | Both | SA | PC | Both |
| Cosmos 14B Agarwal et al. (2025) | **0.374** | **0.268** | **0.212** | 0.331 | **0.243** | 0.182 | **0.091** | **0.069** | **0.020** | **0.333** | **0.241** | **0.184** |
| Cosmos 7B Agarwal et al. (2025) | 0.354 | 0.243 | 0.201 | **0.339** | 0.221 | **0.185** | 0.088 | 0.054 | **0.020** | 0.323 | 0.218 | 0.178 |
| CogVideoX-1.5-5B Hong et al. (2022) | **0.419** | **0.266** | **0.192** | **0.308** | **0.207** | **0.154** | 0.062 | **0.021** | 0.000 | **0.351** | **0.225** | **0.163** |
| CogVideoX-2B Hong et al. (2022) | 0.382 | 0.247 | 0.175 | 0.246 | 0.199 | 0.148 | **0.073** | 0.021 | **0.010** | 0.310 | 0.210 | 0.150 |

Table 9: **We present the evaluation results of 10 video generation models across 10 physics categories.** SA denotes *semantic adherence*, while PC represents *physical commonsense*. **SA, PC** indicates satisfaction of both criteria.

| Model | Fundamental | | | | | | | | | | | | | | | | | | Composite Physics | | | | | | | | | Anti-Physics | | |
|---|---|---|---|---|---|---|---|---|---|---|---|---|---|---|---|---|---|---|---|---|---|---|---|---|---|---|---|---|---|---|
| | C1 | | | C2 | | | C3 | | | C4 | | | C5 | | | C6 | | | C7 | | | C8 | | | C9 | | | C10 | | |
| | SA | PC | Both | SA | PC | Both | SA | PC | Both | SA | PC | Both | SA | PC | Both | SA | PC | Both | SA | PC | Both | SA | PC | Both | SA | PC | Both | SA | PC | b |
| *Proprietary Models* | | | | | | | | | | | | | | | | | | | | | | | | | | | | | | |
| Sora-Turbo | 0.552 | 0.429 | 0.362 | 0.279 | 0.144 | 0.135 | 0.337 | 0.163 | 0.115 | 0.524 | **0.476** | **0.371** | 0.578 | **0.500** | 0.363 | 0.356 | 0.178 | 0.129 | 0.136 | 0.019 | 0.019 | 0.408 | 0.306 | 0.276 | 0.540 | 0.320 | 0.270 | 0.136 | **0.078** | 0.039 |
| Gen-3 | 0.448 | 0.371 | 0.276 | 0.223 | 0.097 | 0.097 | 0.250 | 0.125 | 0.077 | 0.366 | 0.436 | 0.267 | 0.444 | 0.212 | 0.152 | 0.188 | 0.129 | 0.099 | 0.107 | 0.000 | 0.000 | 0.275 | 0.235 | 0.167 | 0.394 | 0.192 | 0.141 | 0.108 | 0.029 | 0.020 |
| Kling-1.6 | 0.510 | 0.471 | 0.356 | 0.272 | 0.136 | 0.136 | 0.352 | 0.171 | 0.143 | 0.515 | 0.455 | 0.347 | 0.571 | 0.418 | 0.297 | 0.283 | 0.141 | 0.130 | 0.196 | 0.022 | 0.022 | 0.369 | 0.243 | 0.184 | 0.370 | 0.259 | 0.210 | 0.125 | 0.073 | **0.042** |
| Pika 2.0 | **0.731** | **0.587** | **0.519** | **0.446** | **0.267** | **0.228** | **0.544** | **0.291** | **0.243** | **0.660** | 0.456 | 0.350 | **0.642** | 0.347 | 0.274 | **0.500** | **0.302** | **0.260** | 0.314 | 0.029 | 0.020 | 0.469 | **0.396** | **0.354** | **0.656** | **0.398** | **0.344** | **0.228** | 0.043 | 0.011 |
| Luma | 0.533 | 0.352 | 0.333 | 0.260 | 0.077 | 0.067 | 0.423 | 0.183 | 0.173 | 0.544 | 0.359 | 0.320 | 0.628 | 0.395 | 0.302 | 0.395 | 0.185 | 0.123 | **0.317** | **0.037** | **0.037** | 0.317 | 0.297 | 0.218 | 0.350 | 0.263 | 0.225 | 0.056 | 0.000 | 0.000 |
| *Open-sourced Models* | | | | | | | | | | | | | | | | | | | | | | | | | | | | | | |
| Hunyuan 720p | 0.505 | **0.447** | **0.350** | 0.340 | 0.150 | 0.130 | 0.333 | 0.167 | 0.137 | 0.462 | 0.413 | 0.279 | 0.350 | 0.272 | 0.155 | 0.323 | 0.219 | 0.177 | 0.196 | 0.000 | 0.000 | **0.436** | **0.416** | **0.327** | 0.396 | **0.375** | 0.271 | 0.082 | 0.021 | 0.010 |
| Open-Sora 2.0 | 0.534 | 0.303 | 0.220 | 0.411 | 0.234 | 0.189 | 0.403 | 0.225 | 0.179 | 0.454 | 0.303 | 0.214 | 0.383 | 0.310 | 0.237 | **0.419** | 0.233 | 0.191 | 0.157 | 0.022 | 0.022 | 0.307 | 0.265 | 0.176 | 0.373 | 0.319 | 0.264 | 0.056 | 0.021 | 0.010 |
| Open-Sora-Plan 1.3 | 0.221 | 0.202 | 0.144 | 0.107 | 0.049 | 0.039 | 0.086 | 0.019 | 0.019 | 0.252 | 0.252 | 0.194 | 0.265 | 0.235 | 0.122 | 0.120 | 0.060 | 0.060 | 0.097 | 0.010 | 0.010 | 0.115 | 0.077 | 0.058 | 0.266 | 0.085 | 0.064 | 0.069 | **0.059** | **0.039** |
| CogVideoX-1.5 | 0.476 | 0.272 | 0.223 | 0.333 | 0.125 | 0.115 | **0.438** | 0.198 | 0.156 | **0.470** | **0.500** | **0.350** | 0.495 | **0.326** | 0.179 | 0.304 | 0.174 | 0.130 | 0.143 | 0.010 | 0.000 | 0.367 | 0.306 | 0.235 | 0.413 | 0.304 | 0.228 | 0.062 | 0.021 | 0.000 |
| Step-video-T2V | 0.401 | 0.265 | 0.204 | 0.273 | 0.202 | 0.166 | 0.361 | 0.239 | 0.145 | 0.371 | 0.264 | 0.227 | 0.384 | 0.273 | 0.238 | 0.328 | 0.227 | 0.154 | 0.165 | 0.000 | 0.000 | 0.388 | 0.385 | 0.257 | **0.446** | 0.368 | **0.316** | **0.085** | 0.021 | 0.020 |
| Wanx-2.1 | 0.422 | 0.281 | 0.235 | **0.355** | **0.255** | **0.205** | 0.374 | **0.255** | **0.195** | 0.413 | 0.298 | 0.214 | 0.386 | 0.264 | **0.249** | 0.384 | **0.273** | **0.222** | **0.244** | **0.022** | **0.022** | 0.310 | 0.313 | 0.266 | 0.415 | 0.370 | 0.273 | **0.085** | 0.021 | 0.020 |
| LTX-Video | 0.363 | 0.176 | 0.167 | 0.152 | 0.010 | 0.010 | 0.191 | 0.032 | 0.021 | 0.212 | 0.202 | 0.121 | 0.268 | 0.144 | 0.113 | 0.138 | 0.050 | 0.050 | 0.092 | 0.000 | 0.000 | 0.135 | 0.104 | 0.042 | 0.305 | 0.110 | 0.085 | 0.076 | 0.011 | 0.000 |

findings suggest that, alongside scaling, dedicating more compute to finer-grained sampling can materially improve adherence to physical laws.

## H    DETAILED PERFORMANCE OVER ALL PHYSICS CATEGORIES

In the main paper, we presented results across different levels of physics. Here, we provide a detailed breakdown of results for each specific physics category in Tab. 9.

## I    LIMITATIONS

While `PhyWorldBench` spans a broad set of physical scenarios with its 1,050 prompts across ten categories, it may not fully capture very specialized or highly intricate interactions that occur in some domains or advanced applications. Similarly, our CAP evaluator generally provides balanced feedback, yet it can display a slight preference for videos with particularly polished visual presentation, such as smooth lighting or dynamic camera movement. Future work will focus on expanding the prompt collection to include additional niche phenomena and on enhancing the evaluation workflow to more clearly distinguish visual style from physical correctness.

## J    CODE AND DATA AVAILABILITY

To ensure full reproducibility, we provide both the generated video assets and all experiment code:

- **Generated videos and prompts:** All videos used in our experiments, along with the corresponding prompt JSON files, are hosted on HuggingFace at `https://huggingface.co/datasets/phyworldbench/phyworldbench`. Note that these generated videos are not part of `PhyWorldBench`, they were instead utilized for our paper experiments. Also, the HuggingFace does not contain the evaluation standards, those are part of the GitHub listed below.

- **Experimentation code and instructions:** The complete implementation of the experiments code as well as detailed setup instructions are available at `https://github.com/ashwin-333/phy-world-bench`.

## K QUERY USED

Here we list the three MLLM queries used in this work, corresponding to the first-stage description prompt (Tab. 10), the second-stage structured-answer prompt (Tab. 11), and the detailed-prompt refinement query (Tab. 12). Note that for proprietary MLLMs such as GPT-o1, GPT-4o and Gemini-2.0-Flash, the model may refuse to evaluate a small fraction of generated videos; the refusal rate is below 3

Table 10: **First prompt for MLLM to generate a detailed description.**

Suppose you are an expert in judging and evaluating the quality of AI-generated videos. These are frames evenly sampled from a generated video from the beginning to the end. This is a generated video from a video model rather than captured from real world, so the video could be low quality, such as fuzzy, inconsistency, especially not following real world physics. Please tell me what is in this video, including what happened and any physics phenomena you observe. Please be sure to include objects in the video, the main event, and any physics phenomena you observe.

Table 11: **Second prompt for MLLM to generate the final answer**

Suppose you are an expert in summarization and finding answers. Here is the text description from another large language model about an AI generated video: "{previous_response}". Based on this description, compare the objects and quantities present in the video with the specified object(s): "{object}". Answer "Yes" if the object(s) could be found in the video, otherwise answer "No". Also, please check if "{event}" is visually depicted in the video, and answer "Yes" or "No". Lastly, please check if video satisfies the standards in list: "{physics_phenomenon_list}", and answer "Yes" or "No" for each standard in the list. Return your evaluation in the following JSON format:
"Objects": "Yes/No",
"Event": "Yes/No",
"Standard_1": "Yes/No",
"Standard_2": "Yes/No",
"...": "Yes/No"

Table 12: **MLLM prompt used to get detailed video generation prompts.**

Refine the sentence: "{prompt}" to contain subject description, action, scene description. (Optional: camera language, light and shadow, atmosphere) and conceive some additional actions to make the sentence more dynamic. Make sure it is a fluent sentence, not nonsense.

## L QUALITATIVE ANALYSIS ON INDIVIDUAL MODEL

We present here qualitative examples from various models, as shown Figure 8, Figure 9, Figure 10, Figure 11, Figure 12, Figure 13, Figure 14, Figure 15, Figure 16, Figure 17, Figure 18, Figure 19 that demonstrate violations of fundamental physics laws, highlighting their limitations. Below, we summarize key observations for each model:

- **CogVideoX-1.5**: Tends to produce distorted objects, such as non-spherical balls that appear dented. Movements are often excessively fast, blurred, erratic, or jittery, sometimes appearing unnaturally sped up.

- **Gen-3**: Generates videos with a cinematic quality, featuring rich lighting and a high level of realism, akin to footage captured with a high-quality camera. Lighting is particularly smooth and well-balanced.

- **Hunyuan**: Produces highly realistic videos, with a quality resembling footage taken by a high-end camera.

- **Kling**: Generates realistic videos, often incorporating dynamic, moving shots that simulate footage captured with a moving camera.

Table 13: Rationalization rate among failure cases. Values represent the percentage of failed videos exhibiting static-scene rationalization.

| Model | Pika | Sora | Kling | Luma | Wanx | Gen-3 | Hunyuan | Step | CogVideo | Open-S | Open-S-P | LTX |
|---|---|---|---|---|---|---|---|---|---|---|---|---|
| Rate | 34% | 24% | 24% | 22% | 28% | 14% | 20% | 18% | 16% | 4% | 6% | 4% |

- **LTX-Video**: Produces grainy footage with distorted objects that are difficult to identify. Moving objects often shift shape and struggle to maintain form.

- **Luma**: Tends to generate realistic footage but with overly vibrant or bright colorization. Moving objects sometimes experience distortion or loss of form.

- **Open-Sora-Plan**: Generates relatively realistic videos but with low fidelity, often lacking detail and featuring still objects. Movement can appear overly smoothed or surreal, sometimes resembling slow-motion. The output often has a VHS-like quality.

- **Open-Sora**: Generates videos where objects frequently distort and warp, failing to hold their shape. The outputs often appear less realistic, resembling animation or 3D graphics.

- **Pika**: Produces realistic yet highly stylized videos. Lighting and objects have an unrealistic softness, contributing to a distinctive stylized aesthetic.

- **Sora**: Generates highly realistic videos with cinematic or stylized lighting. Movement is often exaggerated, enhancing the dramatic quality of the footage.

- **Wanx**: Produces visually crisp footage, yet physical interactions are frequently unreliable; rigid bodies may pass through one another, gravity-driven motions appear "floaty," and impacts are often muted or entirely missing.

- **Step-Video-T2V**: Creates sharp, aesthetically pleasing clips, yet physical reasoning is routinely oversimplified; light fails to refract, flexible objects stay rigid, mass differences do not affect motion, and phenomena such as duplication or buoyancy are either absent or behave contrary to real-world expectations.

## M  STATIC-SCENE RATIONALIZATION AS AN EMERGING FAILURE MODE

Beyond explicit physical violations, we observe that several text-to-video models exhibit a distinct failure mode in which the model generates a partially or fully static scene to avoid producing incorrect physical motion. Instead of attempting the intended physical interaction, the model suppresses motion entirely while still producing a visually coherent video. This behavior effectively "rationalizes" the violation by minimizing dynamics.

To systematically investigate this phenomenon, we analyzed 600 failure cases from our evaluation set, sampling 50 failed videos from each of the 12 models. Each video was annotated for whether it exhibits *static-scene rationalization*, defined as replacing the expected motion with a static or minimally animated scene.

A clear trend emerges: models with higher overall generation quality tend to exhibit substantially higher rationalization rates. Lower-performing models generally fail due to more fundamental issues—such as incorrect object identities, distorted geometry, or inconsistent motion—leaving little opportunity to "rationalize." In contrast, stronger models, once capable of producing semantically aligned and visually coherent scenes, often avoid explicit physics violations by freezing or minimizing motion, thus prioritizing aesthetic quality over physical correctness.

This pattern highlights a key challenge for future text-to-video systems. As models advance in visual fidelity, they may increasingly adopt conservative strategies that circumvent difficult physics rather than attempting them. Quantifying this emerging failure mode is essential for developing models that do not merely avoid errors but actively engage in physically grounded reasoning.

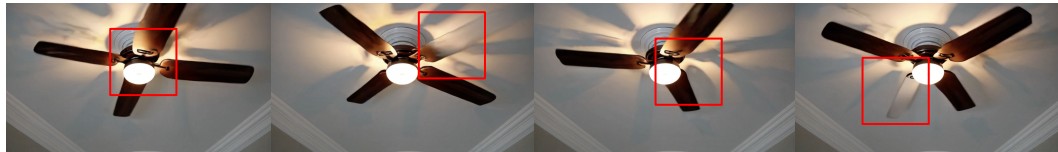

(a) *A ceiling fan spins under a bright light.*

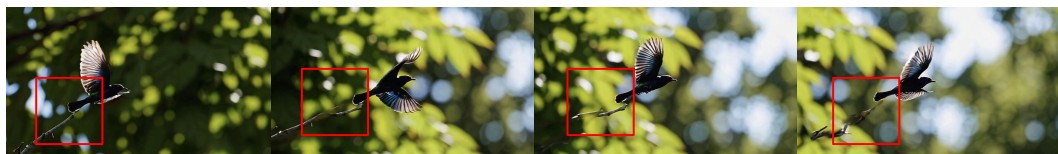

(b) *A bird flaps its wings to take off by generating lift through strong downward wing beats.*

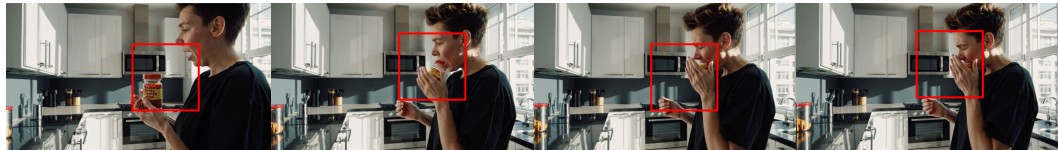

(c) *A person reacts after inhaling pepper.*

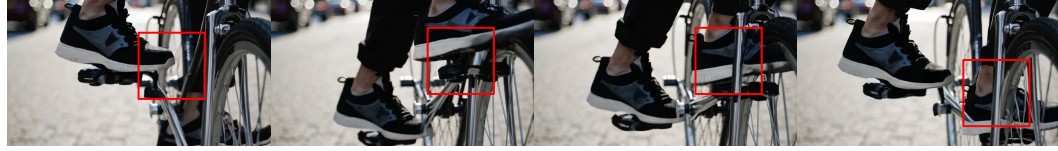

(d) *A person steps on a bicycle's pedal.*

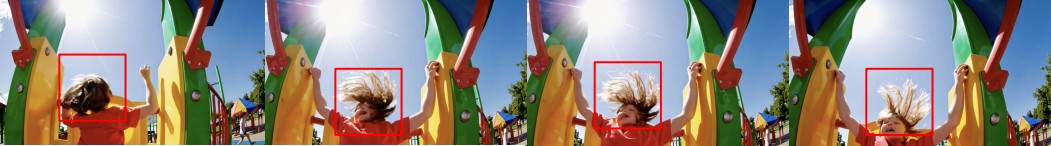

(e) *A child slides down from the top of a slide, speeding up as they descend.*

Figure 8: **Physics violations observed in videos generated by the Sora-Turbo model.** a. Ceiling fan's blades phase into the ceiling during rotation. b. Bird's supporting branch phases into nearby leaves, disconnected from the bird's movement. c. Bottle enters the person's mouth without spatial adjustment as they react to pepper. d. Person pedals through a bicycle instead of interacting correctly with the pedals. e. Child's face deforms unnaturally while sliding down a slide.

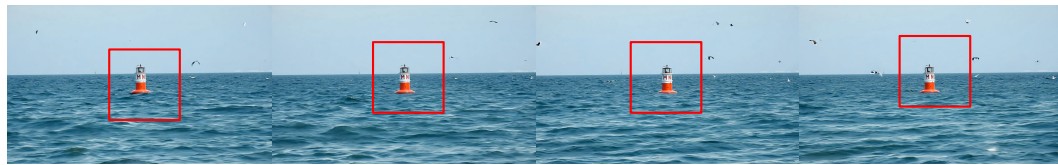

(a) *A buoy is in the ocean.*

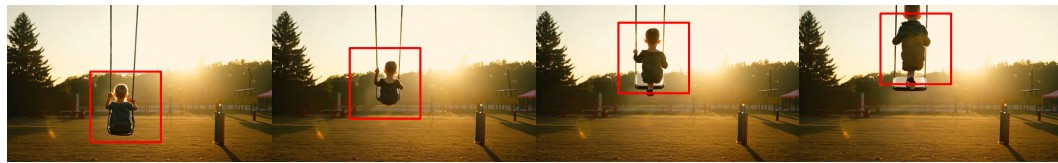

(b) *A child on a swing experiences air resistance.*

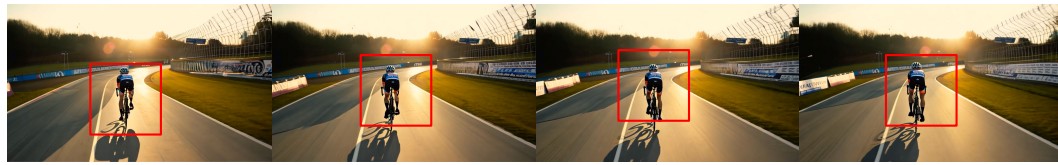

(c) *A cyclist rounds a curve on a track.*

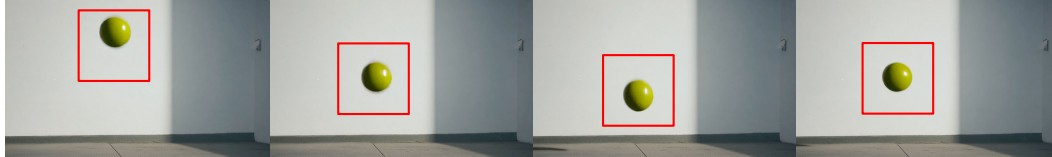

(d) *A rubber ball bounces off a wall.*

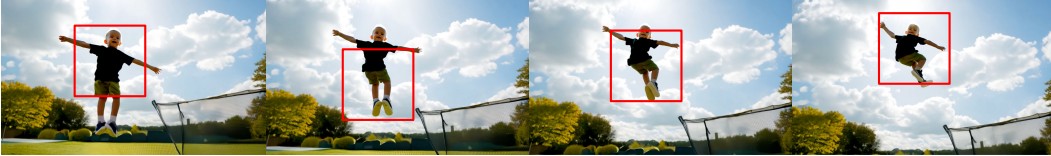

(e) *A child jumps onto a trampoline.*

Figure 9: **Physics Violations Observed in Videos Generated by the Gen-3 Model.** a. Buoy remains unnaturally stationary despite visible water movement. b. Child morphs into different positions and exhibits abnormal motion. c. Cyclist pedaling forward moves backward while rounding a curve on a track. d. Rubber ball bounces off the air instead of the wall. e. Child displays abnormal motion, with knees bending unnaturally.

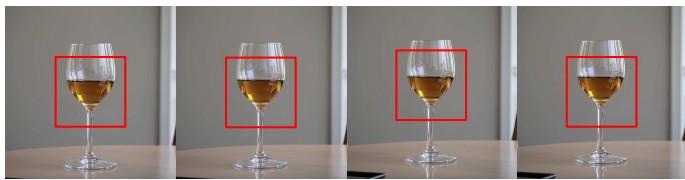

(a) *A glass sits on the table, and the wine in the glass has reversed gravity.*

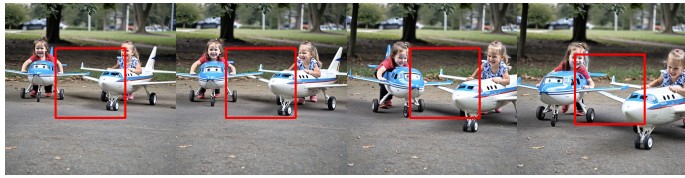

(b) *A small toy airplane and a full-sized airplane are pushed along the ground by two kids.*

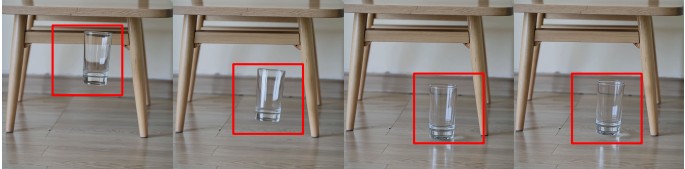

(c) *A glass cup falls from a table.*

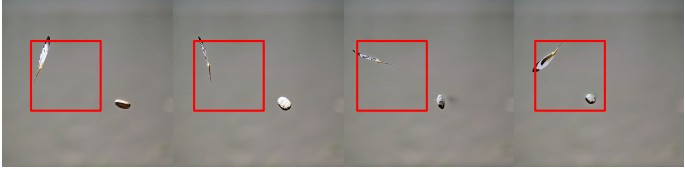

(d) *A feather and a stone are dropped in the*

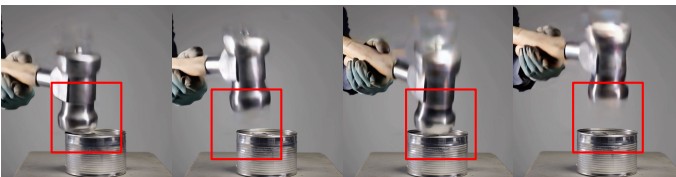

(e) *A hammer strikes an aluminum can.*

Figure 10: **Physics Violations Observed in Videos Generated by the Kling-1.6 Model.** a. Glass with wine remains unaffected by reversed gravity, showing no changes in the liquid's behavior. b. Small toy airplane and full-sized airplane are depicted as the same size, violating realistic proportions. c. Glass cup glides down instead of falling naturally and slides abnormally after hitting the ground. d. Feather and stone fall at the same speed, inaccurately disregarding differences in air resistance. e. Aluminum can remains intact after being struck by a hammer, failing to collapse as expected.

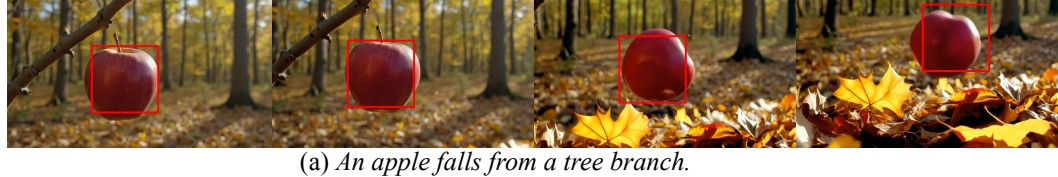

(a) *An apple falls from a tree branch.*

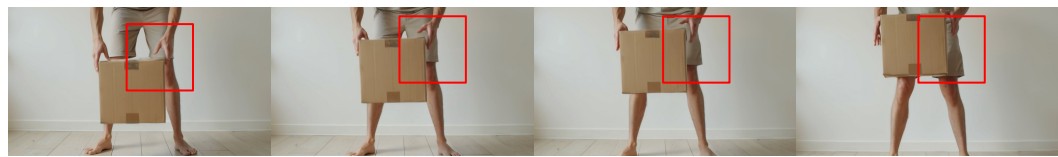

(b) *A person lifts a heavy box from the ground.*

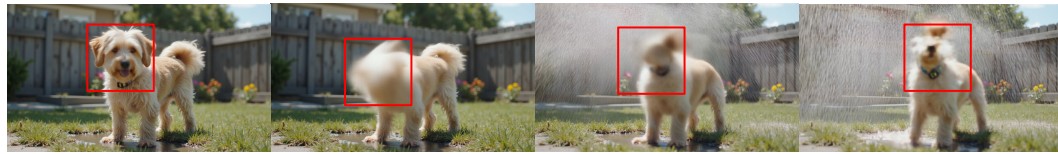

(c) *A dog shakes its body to dry off.*

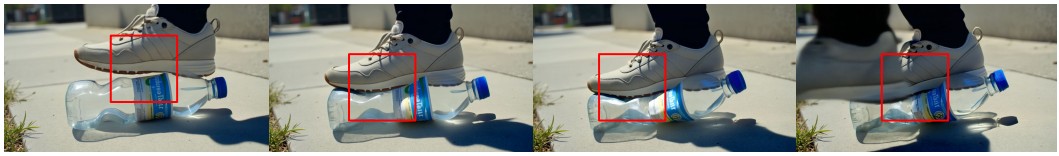

(d) *A plastic bottle is stepped on.*

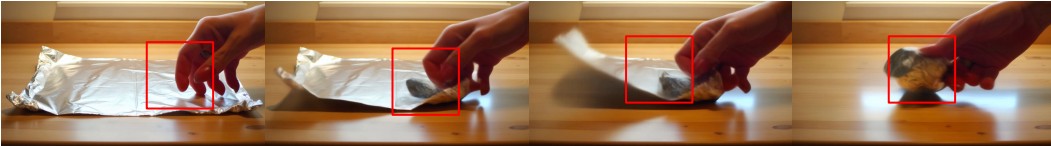

(e) *A piece of tin foil is crumpled.*

Figure 11: **Physics Violations Observed in Videos Generated by the Pika 2.0 Model.** a. Apple floats in the air, then suddenly falls unnaturally as the camera moves. b. Box moves up without any contact from hands. c. Dog's face deforms unnaturally along with water motion as it shakes to dry off. d. Plastic bottle resists fully crushing when stepped on. e. Tin foil folds automatically into a person's hand without proper crumpling motion.

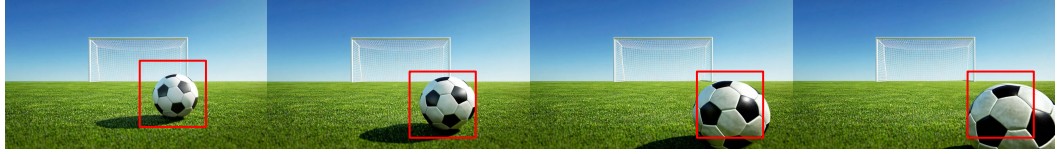

(a) *A soccer ball rolls toward the goal directly after being kicked.*

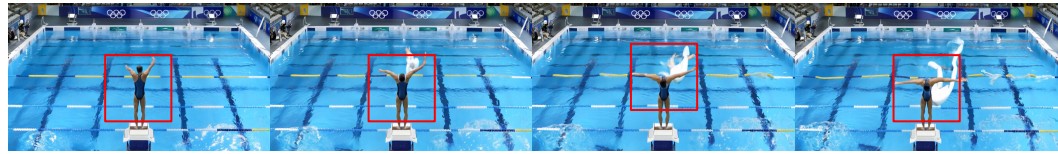

(b) *A diver stands on a platform and jumps into a pool.*

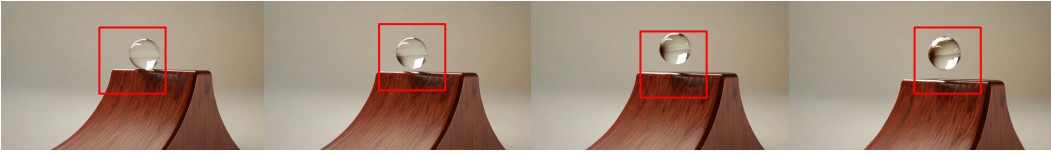

(c) *A marble is released at the top of a half-pipe.*

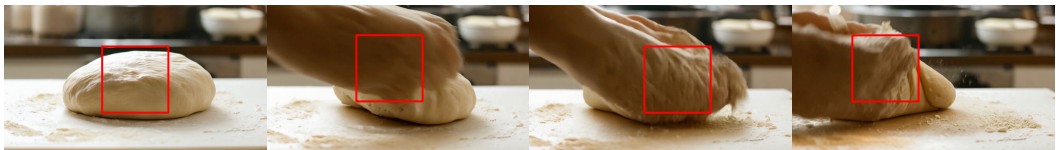

(d) *A lump of dough is stretched out.*

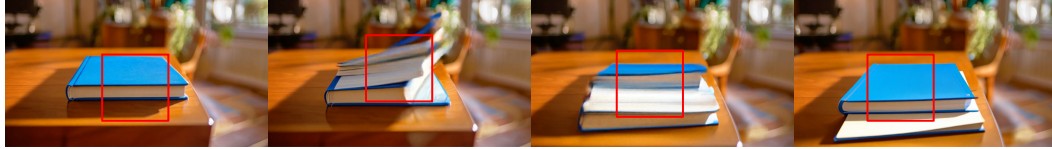

(e) *A book slides off a table.*

Figure 12: **Physics Violations Observed in Videos Generated by the Luma Model.** a. Soccer ball rolls away from the goal in a straight line, showing no natural curve or arc in its path after being kicked. b. Diver exhibits abnormal motion and phases into the water upon jumping from a platform. c. Marble floats unnaturally upward, deviating from expected motion. d. Hands phase into a lump of dough while stretching it out. e. Book exhibits abnormal phasing during its motion.

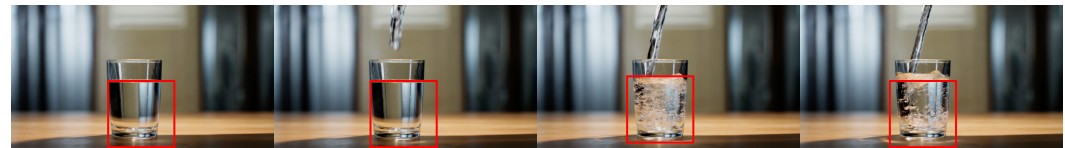

(a) *A glass cup is table-phasing, and it passes through a solid table instead of resting on it.*

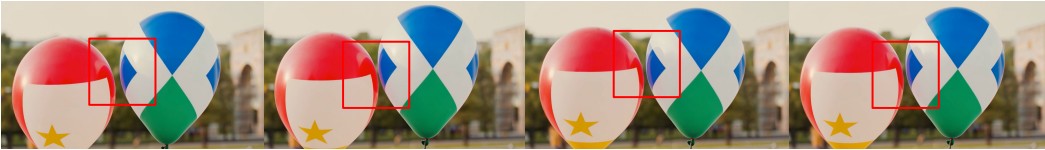

(b) *Two balloons with like charge are placed close together.*

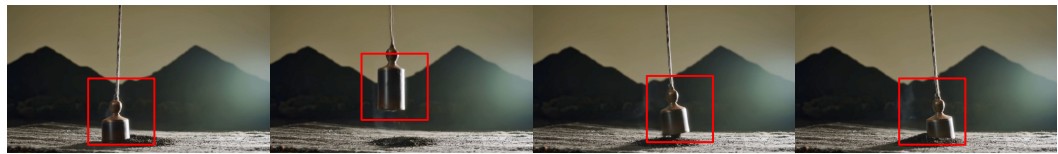

(c) *A pendulum swings between two peaks.*

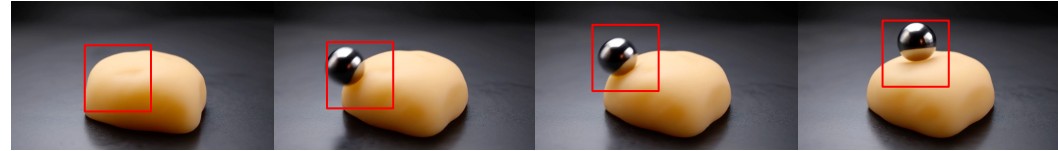

(d) *A steel ball drops onto a soft clay block.*

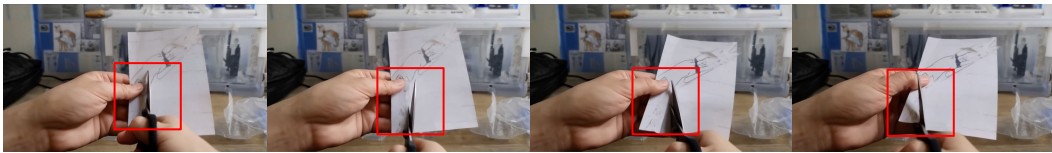

(e) *Scissors cut through paper.*

Figure 13: **Physics Violations Observed in Videos Generated by the Hunyuan 720p Model.** a. Glass cup rests on the table instead of phasing through it, failing to follow the anti-physics prompt. b. Two balloons with like charges remain stationary instead of repelling each other, disobeying electromagnetic interaction physics. c. Pendulum swinging between two peaks exhibits irregular and unrealistic motion. d. Steel ball dropping onto a soft clay block does not deform the clay, defying expected material interaction physics. e. Scissors cutting through paper fail to visibly separate the paper, disobeying shear and cutting physics.

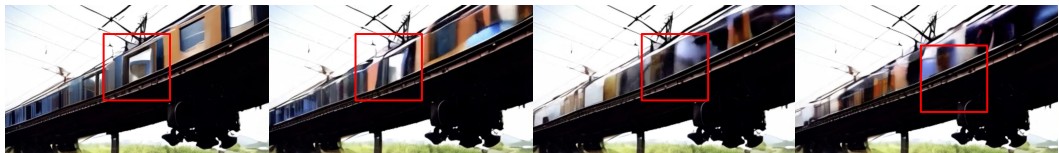

(a) *A train travels upside down, suspended from the tracks above.*

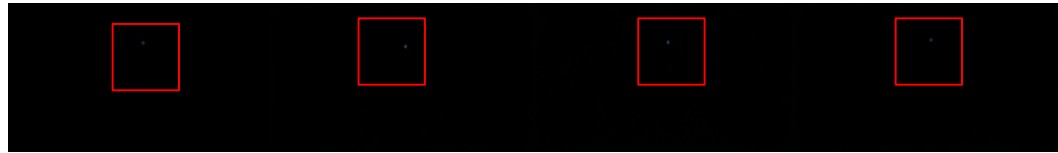

(b) *A flashlight moves across a dark room.*

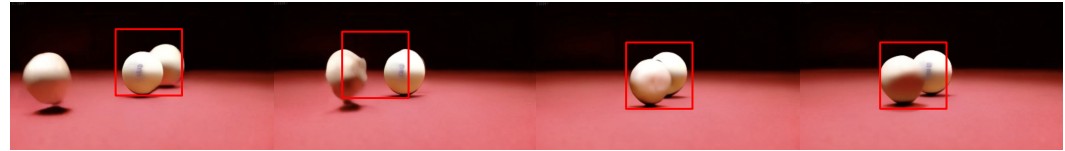

(c) *Two billiard balls collide.*

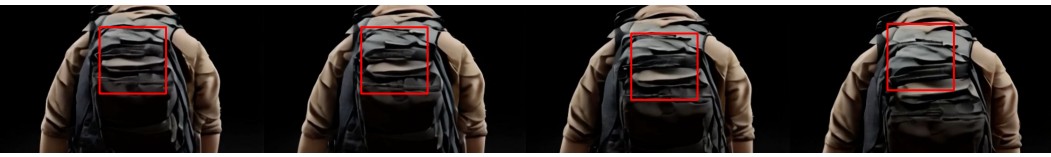

(d) *A backpack is loaded off center.*

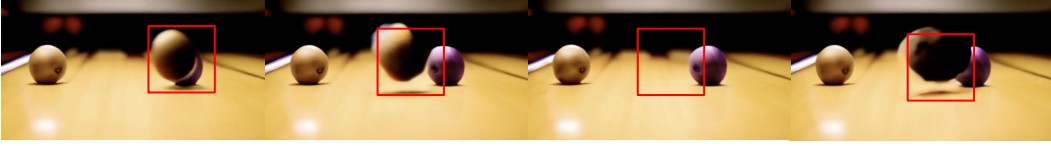

(e) *A bowling ball rolls toward the pins.*

Figure 14: **Physics Violations Observed in Videos Generated by the Open Sora Model.** a. Train traveling upside down on suspended tracks appears right-side-up, failing to depict the intended anti-physics scenario. b. Flashlight moving across a dark room does not illuminate its surroundings. c. Two billiard balls collide but awkwardly phase into each other instead of bouncing off. d. Backpack loaded off-center remains unnaturally straight, with no tipping or imbalance. e. Bowling ball rolling toward pins exhibits abnormal phasing and unnatural motion.

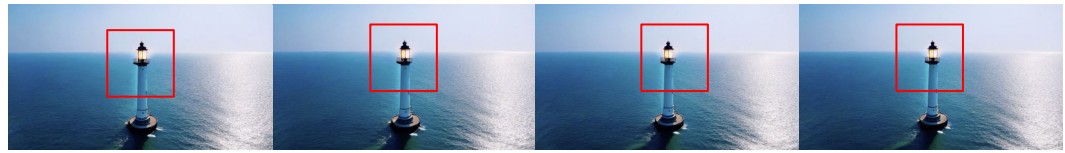

(a) *A lighthouse beam rotates across the ocean.*

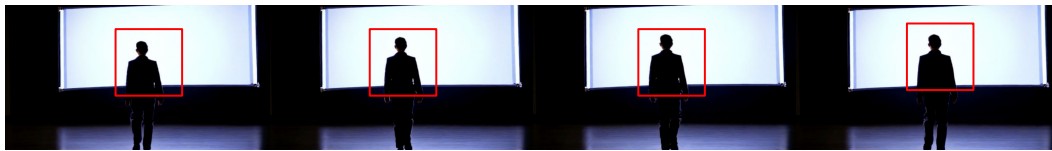

(b) *A person walks in front of a projector screen.*

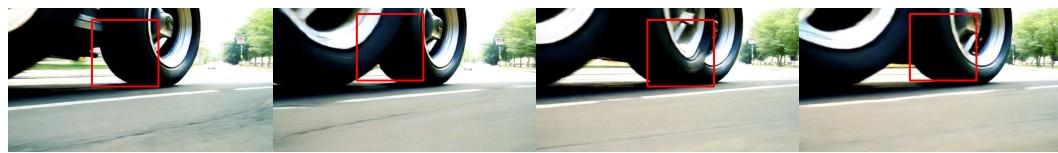

(c) *A rolling tire moves down the street.*

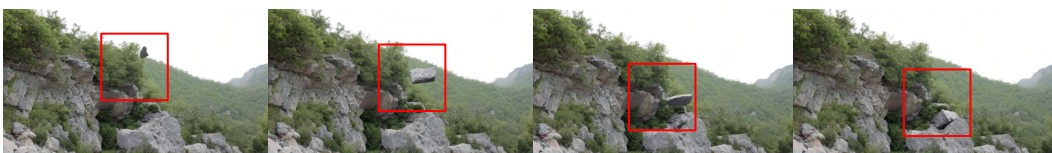

(d) *A rock is dropped from a cliff.*

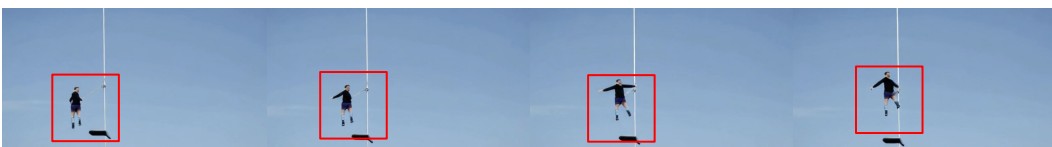

(e) *A tightrope walker moves across a wire.*

Figure 15: **Physics Violations Observed in Videos Generated by the Open-Sora-Plan 1.3 Model.** a. Lighthouse beam fails to visibly rotate across the ocean, undermining its expected motion. b. Person walking in front of a projector screen casts no shadow on the screen. c. Rolling tire phases into another tire while moving down the street. d. Rock dropped from a cliff glides unnaturally and exhibits abnormal motion and phasing. e. Tightrope walker disobeys gravity with abnormal movement while crossing the wire.

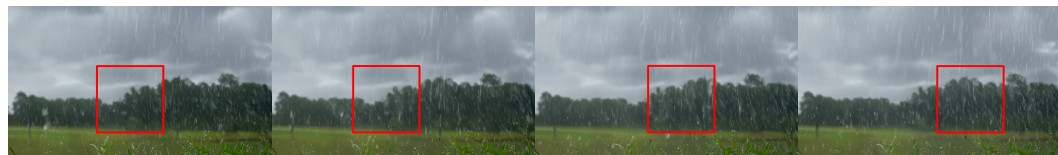

(a) *Raindrops return to the clouds after hitting the ground.*

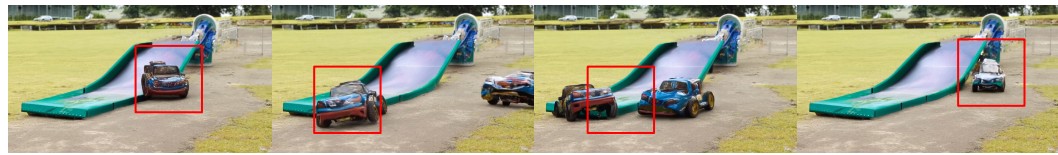

(b) *A toy car and a real car roll down identical ramps.*

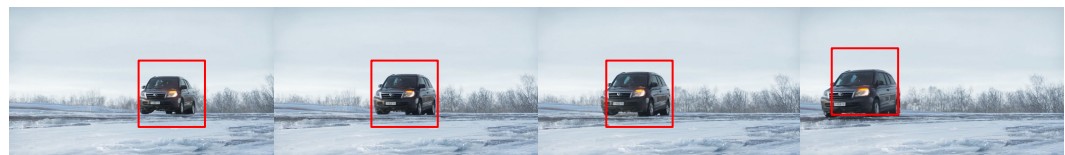

(c) *A car drives on an icy road.*

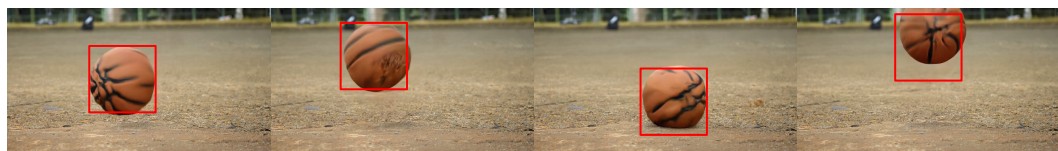

(d) *A basketball is dropped onto the ground.*

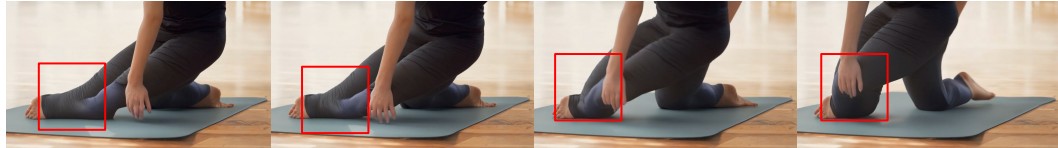

(e) *A person stands on one leg during yoga.*

Figure 16: **Physics Violations Observed in Videos Generated by the CogVideoX-1.5 5b Model.** a. Raindrops do not return to the clouds after hitting the ground; instead, they exhibit normal rainfall, failing to demonstrate anti-physics. b. Toy car and real car rolling down ramps are depicted as the same size, violating proportional realism. c. Car driving on an icy road shows no slippage, unrealistically maintaining normal traction. d. Basketball dropped onto the ground displays unnatural bouncing motion. e. Person standing on one leg during yoga exhibits unrealistic human anatomy.

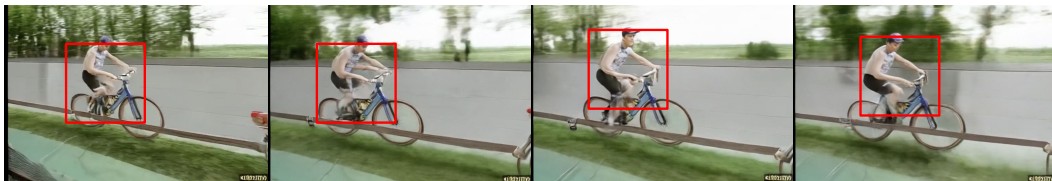

(a) *A stationary bicycle starts pedaling itself without a rider.*

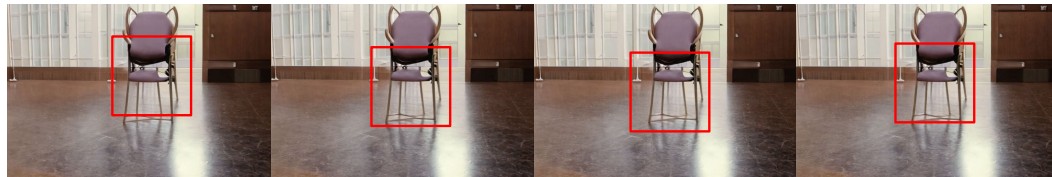

(b) *A chair is placed on an uneven floor.*

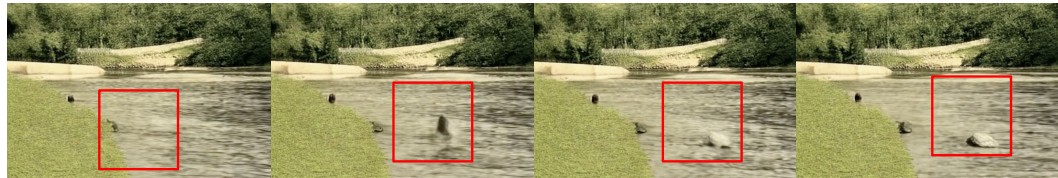

(c) *A stone is thrown across a river.*

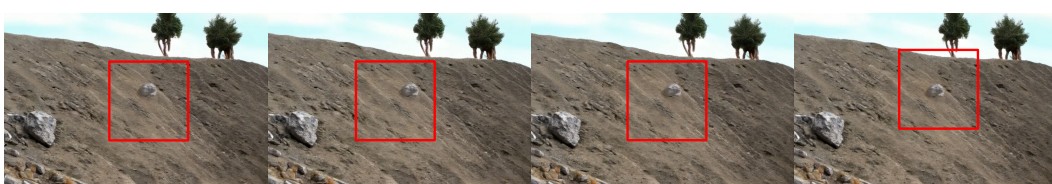

(d) *A rock sits at the edge of a hill, and begins to move.*

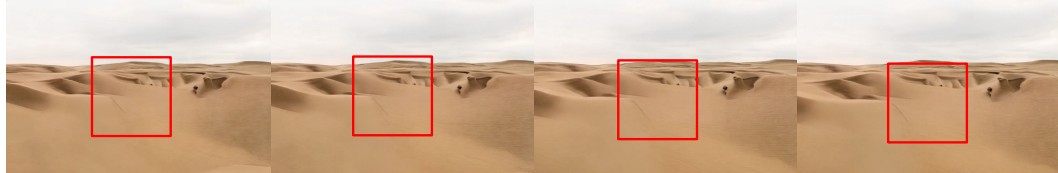

(e) *Wind blows over sand dunes.*

Figure 17: **Physics Violations Observed in Videos Generated by the LTX-Video Model.** a. Stationary bicycle appears with a rider, failing to follow the anti-physics prompt of pedaling without one. b. Chair placed on an uneven floor remains stable, inaccurately disregarding the effect of the uneven surface. c. Stone thrown across a river skips the parabolic motion and instead appears directly in the water. d. Rock sitting on the edge of a hill does not fall, disobeying natural gravitational physics and the prompt. e. Wind blowing over sand dunes fails to pick up sand particles, violating expected particle behavior physics.

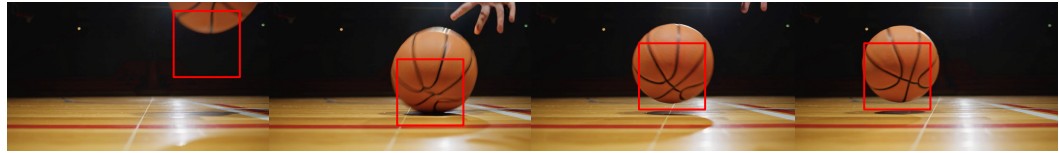

(a) *A basketball hits the ground after being dropped.*

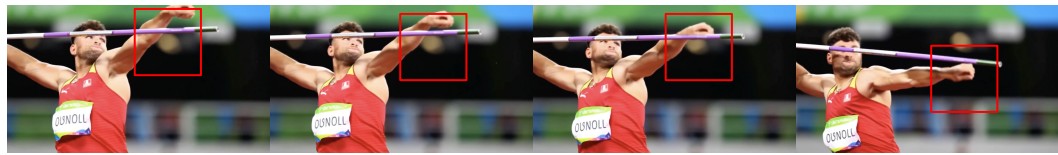

(a) *An Olympic athlete throws a javelin.*

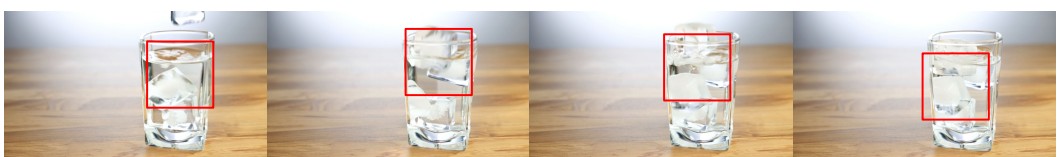

(a) *Ice cubes are placed in a glass of hot water.*

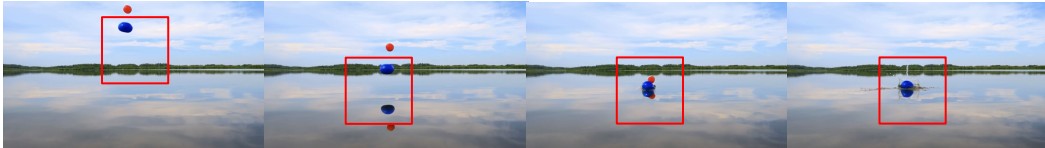

(a) *A stone is dropped into a still pond.*

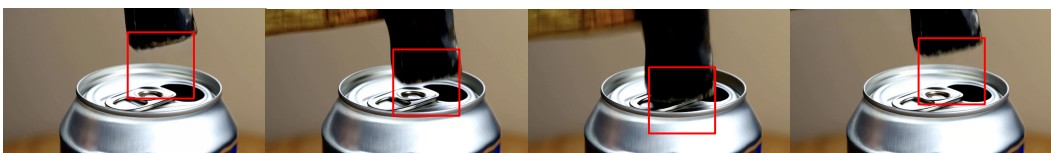

(a) *A hammer strikes an aluminum can.*

Figure 18: **Physics Violations Observed in Videos Generated by the Wanx-2.1 Model.** a. Basketball dropped from shoulder height rebounds to only a fraction of its release height, indicating excessive inelasticity. b. During the javelin throw, the athlete's fore-arm visually interpenetrates the javelin shaft, breaking rigid-body constraints. c. Multiple ice cubes clip through one another instead of exhibiting realistic rigid-body collisions and buoyancy. d. The stone ricochets repeatedly on a calm pond surface and never sinks, contrary to expected gravity-driven submersion. e. The aluminum can remains undeformed when struck by the hammer, ignoring momentum transfer and material plasticity.

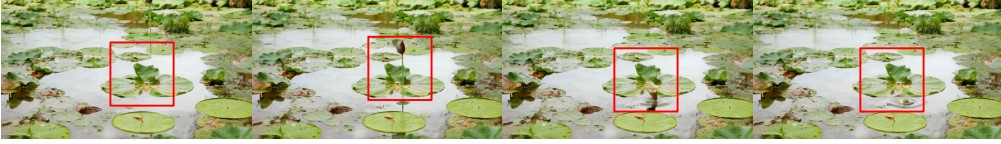

(a) *A beam of light hits the surface of a calm pond, reflecting partially while refracting into the water.*

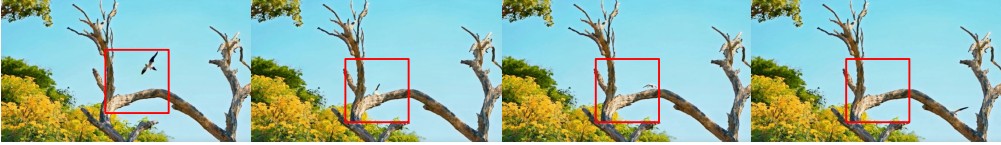

(a) *A tree branch bends in the wind, flexing under pressure and returning upright when the wind subsides.*

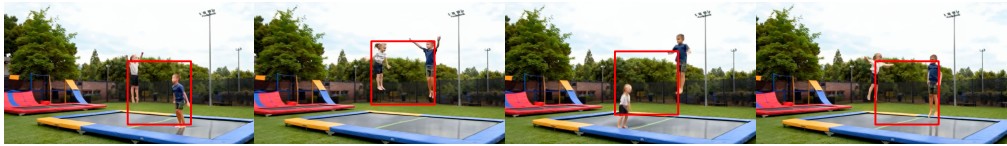

(a) *A child and an adult jump on a trampoline.*

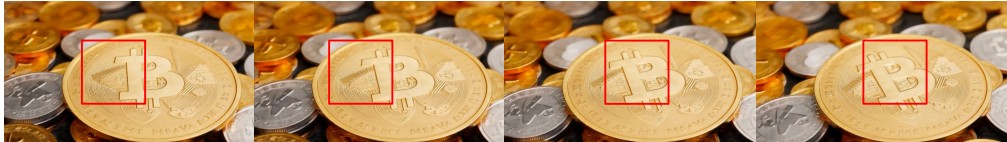

(a) *A single coin continuously splits into two, never stopping.*

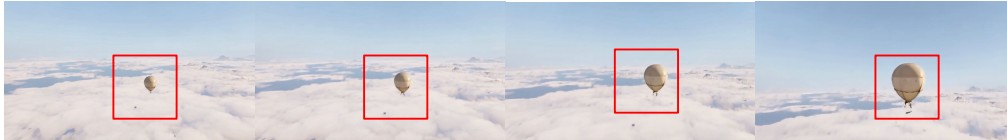

(a) *A balloon has a small hole, causing air to escape from the hole, deflating the balloon and causing it to collapse.*

Figure 19: **Physics Violations Observed in Videos Generated by the Step-Video-T2V Model.** a. An object strikes a calm pond, but there is no visible beam of light or refraction as it enters the water. b. A tree branch remains rigid in the wind, never bending or springing back. c. A child and an adult jump on a trampoline yet rebound to identical heights, ignoring mass-dependent dynamics. d. A single coin was expected to duplicate continuously, but no new coins appear. e. A balloon with a hole should deflate downward; instead, it inexplicably rises, contradicting physics.

Table 14: Video Length by Model

| | CogVideo | Gen-3 | Hunyuan | Kling | LTX-Video | Luma | Open-Sora-Plan | Open-Sora | Pika | Sora | Wanx | Step-Video-T2V |
|---|---|---|---|---|---|---|---|---|---|---|---|---|
| Duration (s) | 5 | 5 | 5 | 5 | 5 | 5 | 5 | 2 | 5 | 5 | 5 | 8 |

## N  VIDEO LENGTH FROM MODELS

## O  FURTHER ANALYSIS ON GENERATIVE MODELS STRUGGLING WITH CERTAIN PHYSICAL SCENARIOS

We identify three key factors behind current models' failures on physics-intensive scenarios: (1) **Training Data Distribution**: large web-scale corpora such as WebVid, Panda70M, or other private web-crawled datasets are dominated by everyday scenes—our analysis shows this yields strong performance on C1 (Object Motion and Kinematics), C4 (Fluid and Particle Dynamics), C5 (Lighting and Shadows), C8 (Rigid Body Dynamics), and C9 (Human and Animal Motion) but underrepresents abstract or multi-object interactions; (2) **Video Captioning Process**: MLLM-based prompt extraction captures main objects and events but routinely omits fine-grained physical dynamics; and (3) **Model Architecture Limitations**: even with detailed prompts, current architectures emphasize short-term frame coherence without physics priors, leading to physically inconsistent outputs—by contrast, models like TORA explicitly model trajectories for better motion fidelity.

## P  LEADERBOARD

Tab. 15 presents the leaderboard performance on both human evaluation and our designed CAP. The performance shows a highly consistency and only difference is that for proprietary models, Sora ranked number 2 while Kling ranked number 2 in CAP. Through further analysis, we found that Kling's more cinematic style can bias CAP toward higher scores despite only moderate physical correctness, whereas our human annotators—who were instructed to focus strictly on physics plausibility—penalize these artifacts, causing Sora to rank higher under human evaluation.

Table 15: **Leaderboard of open-sourced models and proprietary models.** We report leaderboard on both human evaluation and CAP.

| Open models | | Proprietary models | |
|---|---|---|---|
| **Human** | **CAP** | **Human** | **CAP** |
| Wanx | Wanx | Pika 2.0 | Pika 2.0 |
| Hunyuan | Hunyuan | Sora-Turbo | Kling |
| Step-Video-T2V | Step-Video-T2V | Kling | Sora-Turbo |
| Open-Sora | Open-Sora | Luma | Luma |
| CogVideoX-1.5 | CogVideoX-1.5 | Gen-3 | Gen-3 |
| Open-Sora-Plan | Open-Sora-Plan | | |
| LTX-video | LTX-video | | |

## Q  VIDEO QUANTITATIVE ANALYSIS

Here we present the quantitative analysis of the generated video.

Table 16: Quantitative analysis on Erratic Visual Changes.

| Group | Style | Pass / Total | Rate |
|---|---|---|---|
| All models | Abrupt | 53 / 240 | 22.1% |
| | Non-abrupt | 101 / 240 | 42.1% |
| Pika | Abrupt | 6 / 20 | 30.0% |
| | Non-abrupt | 15 / 20 | 75.0% |

Table 17: Quantitative analysis on scene complexity.

| Group | Complexity | Pass / Total | Rate |
|---|---|---|---|
| All models | Complex | 28 / 240 | 11.7% |
| | Less-complex | 44 / 240 | 18.3% |
| Pika | Complex | 3 / 20 | 15.0% |
| | Less-complex | 8 / 20 | 40.0% |

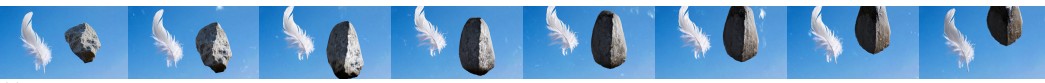

(a) Prompt: A feather and a stone are dropped in the air, where both fall, and the stone drop much faster.

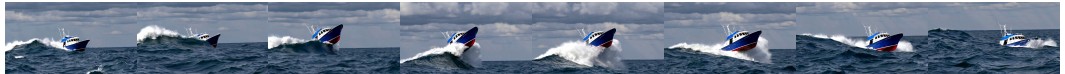

(b) Prompt: A large and small boat float on ocean waves, with the small boat rising and falling dramatically while the larger boat moves steadily with minimal rocking.

Figure 20: **Sora failed examples on multi-object.** We found that generation models tend to fail when the prompt is relatively complex, such as when containing multiple objects.

## R INFLUENCE OF MULTIPLE OBJECTS

We noticed that generation models often fail when the prompt is more complex, such as when containing multiple objects. We show examples of Sora failing when given prompts with multiple objects in Figure 20.

## S DATASET STRUCTURE

The dataset is organized into three main categories of physics phenomena: **Fundamental Physics**, **Composite Physics**, and **Anti-Physics**. Each category is divided into subcategories, which are further broken down into five sub-subcategories, as summarized below:

FUNDAMENTAL PHYSICS

(a) **Object Motion and Kinematics:** Linear Motion, Parabolic Motion, Circular Motion, Rotational Motion, Combined Translational and Rotational Motion

(b) **Interaction Dynamics:** Collisions, Friction and Sliding, Gravity and Free Fall, Electromagnetic Interactions, Tension and Compression

(c) **Energy Conservation:** Potential to Kinetic Energy, Elastic Energy, Energy Loss (Damping), Conservation of Mechanical Energy, Heat Transfer and Phase Changes.

(d) **Fluid and Particle Dynamics:** Water Motion, Smoke and Gas Dynamics, Particle Behavior, Buoyancy and Floating, Viscosity and Flow

(e) **Lighting and Shadows:** Consistency of Illumination, Object Occlusion, Chromatic Aberration and Dispersion, Reflection and Refraction, Shadow Softness and Penumbra

(f) **Deformations and Elasticity:** Material Flexibility, Elastic Rebound, Plastic Deformation, Brittle Fracture, Thermal Expansion and Contraction

COMPOSITE PHYSICS

(a) **Scale and Proportions:** Consistency of Forces Across Scales, Environmental Effects on Different Scales, Proportional Energy Transfer, Structural Integrity Across Scales, Scaling of Mechanical Properties

(b) **Rigid Body Dynamics:** Rotation and Torque, Balance and Stability, Center of Mass, Impact and Deformation, Shear and Cutting

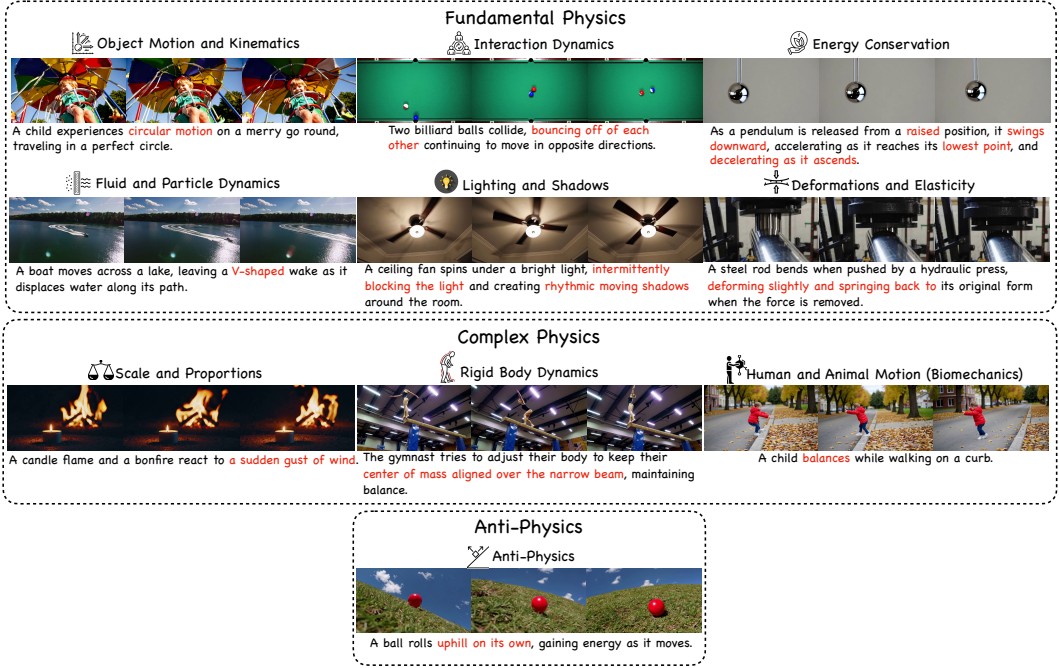

Figure 21: **Dataset Overview.** `PhyWorldBench` contains ten physics categories under three different physics levels.

(c) **Human and Animal Motion:** Anatomical Feasibility, Balance and Posture, Locomotion Across Terrains, Reflexive and Involuntary Motion, Coordination and Fine Motor Skills

ANTI-PHYSICS

(a) **Anti-Physics:** Defying Gravity, Energy Creation or Perpetual Motion, Object Phasing and Surreal Interactions, Time Reversal and Alteration, Infinite Duplication and Division

This hierarchical structure ensures coverage of realistic and non-realistic physical scenarios for diverse analysis. Figure 21 visualises how these ten categories map onto the three physics levels and highlights the numbers of prompts in each branch.

IMPACT STATEMENT

`PhyWorldBench` establishes a new standard for physics-focused text-to-video generation, significantly elevating the field's expectations of realism and accuracy. By rigorously assessing models' capabilities across a comprehensive spectrum of physical phenomena—including motion, energy transfer, and complex interactions—this benchmark not only highlights critical gaps in current systems but also guides researchers toward developing more robust and physics-aware solutions. Moreover, `PhyWorldBench` acknowledges the ethical implications of physics-driven video synthesis, emphasizing the need for transparency, fairness, and responsible deployment. By mitigating risks such as misinformation, biased simulations, and unintended real-world consequences, it ensures that advancements in this field are aligned with scientific integrity and societal well-being. Ultimately, `PhyWorldBench` paves the way for advanced applications in education, scientific visualization, and industry, demanding both photorealism and fidelity to real-world physics while upholding ethical standards.

## T    DATA RELEASE

We publicly released a comprehensive dataset that includes the core assets of `PhyWorldBench`, the 1,050 example json file, the evaluation standards, and the physics categories and subcategories, used to assess physical realism in text-to-video models.

