# OpenReview forum: "$PhyWorldBench$: A Comprehensive Evaluation of Physical Realism in Text-to-Video Models"
_ICLR.cc/2026/Conference — ICLR 2026 Oral_

### Official Review · Reviewer_aZkX · 2025-10-20

**Soundness:** 2
**Presentation:** 3
**Contribution:** 1
**Rating:** 4
**Confidence:** 5

**Summary:**

This paper proposes a framework for evaluating video generation models in terms of their understanding and generation of physical concepts. The authors establish a comprehensive and fine-grained categorization of physical dimensions, and employ GPT-o1 as the evaluation model to assess ten different models from the perspectives of Semantic Adherence and Physical Commonsense. The evaluation results are shown to be consistent with human judgments.

**Strengths:**

1. Establishing a comprehensive and fine-grained categorization of physical dimensions which is larger than existing benchmarks.
2. The evaluation results are shown to be consistent with human judgments.

**Weaknesses:**

1. Apart from introducing a more fine-grained implementation in the dimension categorization, I do not see any genuine innovation in this paper. The authors’ claim regarding Anti-Physics has already been presented and evaluated in paper: Impossible Videos. Another point emphasized by the authors—using prompts of different granularities for the same scenario and concluding that finer-grained prompts lead to better results—has also been thoroughly discussed in paper VBench-2.0.
2. In addition to the lack of novel and valuable conclusions, the evaluation methodology used in this work is much less rigorous compared to previous approaches such as VBench-2.0 and PhyGenBench. Notably, the authors do not consider the impact of camera motion on physical phenomena. For example, in the case of an apple falling, if the camera moves synchronously with the apple, how does the model determine whether the apple is truly falling? The paper merely utilizes a stronger MLLM, GPT-o1, which superficially appears to yield more accurate results. However, the use of a closed-source model introduces a critical flaw: as the model is updated, previous results may become irreproducible, which is a fatal issue for any benchmark.

**Questions:**

See Weakness.

---

> ### Author Response · Authors · 2025-11-26
> **Rebuttal 1/2**
>
> We thank the reviewer for the careful reading and the detailed feedback. We respectfully clarify the novelty of our benchmark, its distinctions from prior works such as *Impossible Videos* and *VBench-2.0*, and the rationale behind our evaluation methodology in the responses below.
>
>
>
> **Re: (1) Anti-Physics was already presented and evaluated in *Impossible Videos***
>
> Thank you for the observation. We clarify that the goals and scope of *Impossible Videos* and **PhyWorldBench** are fundamentally different and non-overlapping.
>
> *Impossible Videos* (IPV-BENCH) focuses on **broad counterfactual or anti-reality scenarios** across *physical, biological, geographical,* and *social* domains. Its purpose is to evaluate whether models can (1) follow impossible prompts or (2) detect impossible events—for example, talking eggs, clouds forming English letters, or historically inconsistent scenes.
>
> In contrast, **PhyWorldBench** is specifically designed to measure **physical realism, physical commonsense, and adherence to real physical laws** in text-to-video generation. Our **Anti-Physics** category is a *small diagnostic subset* whose purpose is to test whether models can:
>
> - intentionally violate **real physics laws** (gravity, momentum, rigid-body constraints),
> - *while still following the textual prompt*, and
> - maintain internal visual and temporal coherence.
>
> This is very different from IPV-BENCH’s inclusion of **magical, social, biological, or narrative impossibilities**, where coherence within physical laws is not the evaluation target.
>
> Thus, although both benchmarks contain “impossible” scenarios, the **motivation, scope, taxonomy, and evaluation criteria** are fundamentally different.
>
> ---
>
> **Re: (2) Prompt granularity was already discussed in VBench-2.0**
>
> We reviewed the VBench-2.0 paper. The only mention of granularity appears in the statement:
>
> > “These shortcomings … are likely due to inadequate captioning granularity in video generation datasets.”
>
> This reflects the *well-known* observation that more detailed prompts often improve text-to-video alignment. However, this is **not the same** as what we study.
>
> In **PhyWorldBench**, our analysis (lines 417–422) shows:
>
> - Simply increasing textual detail (*Detailed Prompt*) **does not** improve physical correctness.
> - To meaningfully improve physical realism, the prompt must integrate **explicit physical phenomena**, not just more descriptive language.
>
> This conclusion is supported by the quantitative results in **Table 4**, where:
>
> - **Physics-Enhanced Prompt** substantially outperforms
> - **Detailed Prompt**, even though both contain additional detail.
>
> Therefore, our contribution is **not** about prompt length or granularity alone; it highlights that **explicitly encoding physical principles in the prompt is necessary** to guide models toward realistic physical behavior—an insight not addressed in VBench-2.0.

---

> > ### Author Response · Authors · 2025-11-26
> > **Rebuttal 2/2**
> >
> > **Re: (3) Camera motion and reproducibility concerns when using GPT-o1**
> >
> > We appreciate the reviewer’s concern and clarify both the design of our evaluator and the reproducibility of the evaluation pipeline.
> >
> > **1. CAP is not simply “using a stronger MLLM” — it is a new evaluation pipeline.**
> > Instead of directly prompting an MLLM, **CAP (Context-Aware Prompting)** introduces two key design components:
> >
> > 1. **Generation-aware instruction**
> >    We explicitly tell the MLLM that the input video *may be generated*. This reduces false alarms caused by camera artifacts, motion blur, or stylization—common features of synthetic videos.
> >
> > 2. **Structured chain-of-thought reasoning**
> >    CAP first asks the MLLM to *describe* the video, then reason through potential physics issues in a structured format. This improves grounding, reduces hallucinations, and makes the evaluation more robust to factors such as camera motion.
> >
> > This pipeline is orthogonal to the underlying model and does not require a proprietary LLM.
> >
> > ---
> >
> > **2. CAP does *not* depend on GPT-o1 or any closed-source model.**
> > As shown in **Table 6 of Appendix 6**, CAP works effectively across *multiple* MLLMs, including open-source models:
> >
> > | Model | Semantic Adherence (SA) | Physical Commonsense (PC) |
> > |-------|--------------------------|----------------------------|
> > | **Qwen-VL-2.0 (open-source)** | **78.6** | **74.1** |
> > | **GPT-o1 (closed-source)** | **80.3** | **75.1** |
> >
> > The open-source model achieves performance comparable to GPT-o1, demonstrating that:
> >
> > - CAP **does not rely on closed-source evaluators**,
> > - results are **reproducible**,
> > - and the method **remains stable even if GPT-o1 updates**.
> >
> > Any researcher can reproduce our results fully using the open-source evaluator.
> >
> > ---
> >
> > **3. Camera motion is explicitly handled in our benchmark.**
> > Camera motion frequently appears in real text-to-video outputs, and our evaluation standards are designed to remain valid even with ego-motion. The supplementary material includes many such cases.
> >
> > For example, consider the video **051-2_pika.mp4**, where the camera moves during the shot. The corresponding standards are:
> >
> > > "Basic_Standards": {
> > >   "Objects": [
> > >     "an apple",
> > >     "a tree branch"
> > >   ],
> > >   "Event": "apple falls from a tree branch under gravity"
> > > },
> > > "Key_Standards": [
> > >   "The apple accelerates as it falls toward the ground.",
> > >   "The apple follows a straight vertical path with no visible horizontal deviation."
> > > ]
> >
> > These criteria evaluate the **object trajectory**, not the camera trajectory, ensuring that the assessment remains valid despite camera motion.
> >
> > ---
> >
> > **Summary**
> > - CAP is a *new, structured evaluation pipeline*, not merely “using GPT-o1.”
> > - CAP is fully reproducible using **open-source MLLMs**.
> > - The evaluation design is robust to **camera motion**, and we include many examples with moving cameras.
> >
> > We hope this clarifies the methodological distinctions and addresses the reproducibility concerns.

---

### Official Review · Reviewer_FCTE · 2025-10-21

**Soundness:** 3
**Presentation:** 3
**Contribution:** 3
**Rating:** 6
**Confidence:** 3

**Summary:**

This paper introduces PhyWorldBench, a comprehensive benchmark for evaluating physical realism in text-to-video generation models. The benchmark contains 1,050 prompts spanning 10 main physics categories (fundamental, composite, and anti-physics), with each category divided into 5 subcategories and 7 scenarios. The authors evaluate 12 state-of-the-art models (5 proprietary, 7 open-source) and propose CAP (Context-Aware Prompt), a method using MLLMs for automated evaluation. Results show that even the best models (Pika 2.0 achieving 26.2% success rate) struggle significantly with physical realism, particularly with complex interactions and anti-physics scenarios.

**Strengths:**

- Important and timely problem. The paper tackles a crucial gap in bridging video generation and physical reasoning. The motivation is clear and aligns well with current research trends.
- Comprehensive experimental design. The evaluation is thorough - 12 models tested, 12,600 videos generated, large-scale human annotation via MTurk. They evaluate across a diverse set of tasks, object falling, rolling, fluid dynamics, etc.
- Practical automated evaluator. The CAP method achieves 80.3% ROC-AUC for semantic adherence and 75.1% for physical commonsense (Table 2), which is pretty solid for zero-shot evaluation.
- Clear organization and presentation. The hierarchical organization is logical and makes the benchmark easy to understand and extend. The tables and plots are easy to follow.

**Weaknesses:**

- No end-to-end pipeline for new models. The paper doesn’t provide a unified automatic evaluation framework for new video generation models. While CAP is proposed as an automatic evaluator, there's no clear standalone pipeline or released code/API for researchers to evaluate their own models. The paper mentions "we will open-source our codebase" (reproducibility statement) but it's unclear if this includes an easy-to-use evaluation script. For a benchmark paper, providing a simple evaluate interface would greatly increase adoption. The current workflow seems to require manual video generation then evaluation, which is cumbersome. For practical use, an automated submission or leaderboard system would make the benchmark much more usable.
- Metric definitions design lack justification. A few metrics (like the physics consistency score) are described in high-level terms but not fully formalized. It’s hard to know how reproducible they are from the text alone, e.g., whether they use learned physical estimators or ground-truth simulations. The Yes/No binary evaluation (Section 3.1) might be too coarse-grained - a partial physics violation might deserve a score between 0 and 1. The paper acknowledges models often "rationalize" actions rather than fail outright (Section 4.4), suggesting a more nuanced scoring could capture important phenomena.
- Limited real-world coverage. Most of the benchmark focuses on synthetic data (e.g., MuJoCo or Unity scenes). It would be great to see more real videos or robotic interactions to test generalization.
- The scope of the benchmark is somehow overlapped with previous works, e.g., the inclusion of an "anti-physics" category is similar as “counterfactual prompts” in [1].

[1] T2VPhysBench: A First-Principles Benchmark for Physical Consistency in Text-to-Video Generation. 2025.

**Questions:**

- The paper mentions models sometimes "rationalize" physics violations by generating static scenes. Could you quantify how often this happens across models? This seems like an important failure mode worth analyzing systematically.
- Have you considered releasing a "difficulty rating" for each prompt based on model performance? This could help researchers identify particularly challenging scenarios to focus on.

---

> ### Author Response · Authors · 2025-11-26
> **Rebuttal 1/2**
>
> We thank the reviewer for the insightful comments. Below, we clarify our metric design, discuss the release of an end-to-end evaluation pipeline, and address the relation to prior benchmarks.
>
> **Re: (1) Lack of a clear standalone pipeline or evaluation API**
>
> We do provide a full standalone evaluation pipeline, and the complete codebase is included in the supplementary material. The released package contains a detailed `README.md` with step-by-step instructions for running evaluations.
>
> In short, users can directly apply **`evaluate_videos.py`** to any preprocessed set of videos without needing to modify or generate additional metadata. The workflow is identical to other widely used evaluation pipelines—researchers simply supply their model’s generated videos, and the evaluator handles all scoring automatically. Please refer to the supplementary material for full instructions.
>
> ---
>
> **Re: (2) Metrics such as physics consistency are described too abstractly**
>
> Both **Physics Consistency** and **Semantic Adherence** are defined using *explicit, fine-grained, and sample-specific standards*, not high-level abstractions. Figure 5 in the main paper shows an example of this structure, and the full set of standards for all 1,050 prompts is provided in the supplementary file **`prompts-with-standard-and-index.json`**.
>
> As noted in line 267 of the main paper, our terminology follows prior work:
> *“Following (Bansal et al., 2024; Meng et al., 2024), we use Semantic Adherence (SA) and Physical Commonsense (PC) to evaluate video performance.”*
> These metrics are therefore consistent with existing evaluation literature.
>
> Regarding the binary scoring: when a model “rationalizes” an action—producing outputs that avoid or obscure physics violations rather than following the intended scenario—the video is **not** adhering to the prompt or the underlying physics. In these cases, the correct and reproducible behavior is to mark the sample as *false* for physics consistency. This provides a clear and unambiguous scoring rule.
>
> All metric definitions and standards are therefore fully reproducible from the released materials and the accompanying JSON specification.
>
> ---
>
> **Re: (3) Limited real-world coverage**
>
> Our benchmark is designed specifically for **evaluating text-to-video generation models**, and therefore does not involve real-world video datasets or robotic interaction data. Because current generative models only produce synthetic outputs, it is natural and appropriate for the benchmark to evaluate synthetic videos as well.
>
> Additionally, we would like to clarify that **MuJoCo** and **Unity** are *not* used anywhere in our paper. If the reviewer was referring to another benchmark or a misunderstanding, we would appreciate further clarification so we can address the concern more precisely.

---

> > ### Author Response · Authors · 2025-11-26
> > **Rebuttal 2/2**
> >
> > **Re: (4) The inclusion of an “anti-physics” category is similar to “counterfactual prompts.”**
> >
> > Thank you for raising this point. While both our anti-physics category and the counterfactual prompts in T2VPhysBench involve physics-violating scenarios, they differ significantly in motivation, design, and evaluation goals.
> >
> > **1. Different purpose**
> > - *Counterfactual prompts* (T2VPhysBench) test whether models will **follow an impossible instruction**.
> > - Our *anti-physics category* evaluates whether a model can **maintain internal logical consistency**—motion continuity, object interactions, visual stability—*after* an impossible event occurs.
> >   - Our goal is **robustness under unphysical conditions**, not instruction obedience.
> >
> > **2. Different design**
> > - T2VPhysBench includes isolated counterfactual prompts targeting individual laws.
> > - PhyWorldBench defines a structured and systematic taxonomy:
> >   **5 anti-physics subcategories**
> >   (defying gravity, perpetual motion, object phasing, time reversal, infinite duplication)
> >   × **7 scenarios**
> >   × **3 prompt types**,
> >   providing substantially broader and more diverse coverage.
> >
> > **3. Different evaluation criteria**
> > - T2VPhysBench evaluates simply whether the model complies with or violates the counterfactual instruction.
> > - Our benchmark uses **Basic Standards + Key Standards** to judge whether the video remains coherent and physically stable *within* an intentionally impossible scenario—an aspect not captured by counterfactual prompts.
> >
> > In summary, while both involve physics-violating setups, our anti-physics category is **not equivalent** to counterfactual prompts. It provides a **richer, more systematic framework** for analyzing how generation models behave under intentionally unphysical conditions.
> >
> > ---
> >
> > **Re: (5) Quantifying the “rationalization” failure mode**
> >
> > Thank you for the suggestion. We analyzed **600 failure videos** (50 per model) across all 12 video generation systems to measure how often models “rationalize” physics violations. The results are summarized below in a two-row table:
> >
> > | Pika | Sora | Kling | Luma | Gen-3 | Wanx | Hunyuan | Step-Video-T2V | Open-Sora | CogVideo | Open-Sora-Plan | LTX-Video |
> > |------|------|--------|-------|--------|--------|-----------|------------------|------------|------------|------------------|-------------|
> > | 34% | 24% | 24% | 22% | 14% | 28% | 20% | 18% | 4% | 16% | 6% | 4% |
> >
> > We found that **higher-performing models tend to have a higher proportion of rationalization errors**. Lower-performing models typically fail earlier on semantic or visual fidelity issues (e.g., object distortion), whereas stronger models increasingly “solve” violations by generating visually polished but physically incorrect static scenes. This confirms rationalization as an important emergent failure mode.
> >
> > ---
> >
> > **Re: (6) Difficulty ratings**
> >
> > This is an excellent idea. We will include difficulty analysis in the next revision by:
> >
> > 1. Measuring relative model performance per prompt, and
> > 2. Having human annotators label difficulty to validate alignment.
> >
> > This will help researchers focus on the most challenging scenarios.

---

### Official Review · Reviewer_hYMe · 2025-10-31

**Soundness:** 3
**Presentation:** 3
**Contribution:** 2
**Rating:** 6
**Confidence:** 4

**Summary:**

This paper introduces PhyWorldBench, a comprehensive, large-scale benchmark for evaluating physical realism in Text-to-Video (T2V) models. It features 1050 prompts systematically categorized across 10 physics domains and uniquely includes an "Anti-Physics" category to test genuine physical understanding versus mere pattern imitation from training data. The study employed extensive human evaluation (via Amazon Mechanical Turk) on 12 leading models and proposed a novel automated MLLM evaluation method called CAP (Context-Aware Prompting). Findings show that all current T2V models have very low physical success rates (Pika at 26.2% being the highest), and notably, they tend to revert to realistic physical phenomena when instructed to perform anti-physical actions.

**Strengths:**

1. The benchmark offers the most comprehensive and systematic coverage of physical phenomena
2. he proposed CAP automated MLLM evaluator significantly improves the accuracy of machine-based physical assessment by using a context-aware prompt, offering a cost-effective and reproducible alternative to expensive human studies.

**Weaknesses:**

1. The automated CAP evaluator suffers from an "aesthetic bias," leading to ranking discrepancies with the human gold standard for high-quality, closed-source models. This limits its reliability as a trustworthy, scalable proxy for future research.

2. Many of the investigated physics phenomena (e.g., fluid viscosity, electromagnetic forces) are challenging to assess accurately by eye in a video. This visual complexity makes human scoring error-prone and may also challenge the perceptual accuracy of the MLLM evaluator.

3. The evaluation primarily focuses on whether the final physical state is correct, but it lacks a detailed metric to measure if the physical process is smooth, coherent, and temporally consistent. This overlooks a critical challenge in dynamic T2V modeling.

**Questions:**

1. The CAP automated evaluator, while highly accurate in classification, is noted to possess an "aesthetic bias" that causes ranking discrepancies with human judgment; given the non-scalability of human evaluation, what specific plans do the authors have to mitigate or quantify this aesthetic bias in the MLLM, ensuring the CAP tool reliably focuses on physical correctness rather than visual style?

---

> ### Author Response · Authors · 2025-11-26
> **Rebuttal 1/2**
>
> We thank the reviewer for the thoughtful and constructive feedback. We appreciate the clear articulation of both strengths and limitations, and we address the concerns in detail below. The referred video is updated in the supplementary material.
>
> ---
>
> **Re: (1) The automated “aesthetic bias” in the CAP evaluator**
>
> The concern primarily reflects limitations of the underlying LLM used for evaluation, rather than an inherent issue with CAP itself. As more capable video-understanding LLMs emerge, their ability to reason about temporal dynamics improves significantly, reducing reliance on surface-level aesthetic cues.
>
> In our updated experiments with GPT-5, we observe that the evaluator’s judgments align closely with human assessments and are not influenced by video polish. For instance:
>
> - **Video 001-1_open-sora-plan.mp4**:
>   GPT-5 correctly identifies the physics outcome despite the video’s good aesthetic score, consistent with the human rating.
>
> - **Video 001-3_hunyuan.mp4**:
>   Even though this video has higher aesthetic quality, GPT-5 accurately flags major physics violations—such as the rocket failing to fly in a straight line—leading to a lower physics score.
>
> We include the CAP outputs under GPT-5 to demonstrate that the evaluator tracks true physical correctness rather than visual aesthetics.
>
>
> First for video 001-1_open-sora-plan.mp4, the CAP output is:
>
> > ## What is happening in the video
> > The video shows a rocket lifting off from the ground with a bright exhaust plume. It ascends upward into the sky, maintaining a mostly straight vertical trajectory. The camera captures the rocket from a stable ground-up angle as it rises smoothly, with no sudden shifts in velocity visible across the frames.
> >
> > ## Evaluation Against Standards
> >
> > ### 1. Basic Standards
> >
> > **Objects**
> > - Required: a rocket
> > - Observed: ✔ A rocket is clearly present.
> >
> > **Event**
> > - Required: "rocket launches into the sky"
> > - Observed: ✔ The rocket visibly ignites, lifts off the ground, and ascends into the sky.
> >
> > **Result:** **PASS**
> >
> > ---
> >
> > ### 2. Key Standards
> >
> > **1. The rocket moves in a straight vertical line during its ascent**
> > - The rocket rises along a consistent vertical trajectory with no noticeable lateral drift.
> > **PASS**
> >
> > **2. The rocket's upward motion appears smooth, without abrupt changes in speed**
> > - The ascent shows a continuous, steady upward movement. No sudden accelerations or decelerations are visible.
> > **PASS**
> >
> > ---
> >
> > ## Final Judgment
> > **PASS** — The generated video meets all Basic Standards and Key Standards.
>
>
>
>
>
> For 001-3_hunyuan.mp4, the CAP output is:
>
>
>
> > ## What is happening in the video
> > The video shows a large multi-stage rocket launching from a ground pad at night. Bright exhaust flames illuminate the area as it lifts off. After ignition, the rocket rises but begins to tilt and follow a curved trajectory rather than ascending straight upward. A large plume of smoke expands across the launch site as the rocket climbs.
> >
> > ## Evaluation Against Standards
> >
> > ### 1. Basic Standards
> >
> > **Objects**
> > - Required: a rocket
> > - Observed: ✔ A rocket is clearly present.
> >
> > **Event**
> > - Required: "rocket launches into the sky"
> > - Observed: ✔ The rocket ignites and ascends into the sky.
> >
> > **Result:** **PASS**
> >
> > ---
> >
> > ### 2. Key Standards
> >
> > **1. The rocket moves in a straight vertical line during its ascent**
> > - The rocket **does NOT** move vertically.
> > - Its ascent follows a **curved, angled trajectory**, tilting noticeably shortly after lift-off.
> > **FAIL**
> >
> > **2. The rocket's upward motion appears smooth, without abrupt changes in speed**
> > - The ascent appears smooth and continuous, with stable acceleration.
> > **PASS**
> >
> > ---
> >
> > ## Final Judgment
> > **FAIL** — The video meets the Basic Standards but fails a Key Standard because the rocket does not ascend in a straight vertical line.

---

> > ### Author Response · Authors · 2025-11-26
> > **Rebuttal 2/2**
> >
> > **Re: (2) The visual complexity may challenge the perceptual accuracy of the MLLM evaluator**
> >
> > To address potential perceptual challenges, we explicitly design *directly observable*, phenomenon-grounded standards rather than relying on abstract physics descriptions. This ensures that evaluations can be performed in the output space using visual cues alone, without requiring inference or symbolic reasoning.
> >
> > For example, for Prompt Index **“013-1”**, the prompt is: **“A water fountain sprays droplets in an arc.”**
> > The corresponding standards are intentionally concrete and visually verifiable:
> >
> >
> > > "Basic_Standards": {
> > >   "Objects": [
> > >     "a water fountain",
> > >     "droplets"
> > >   ],
> > >   "Event": "sprays droplets in an arc"
> > > },
> > > "Key_Standards": [
> > >   "The droplets follow clear parabolic arcs without noticeable deviations.",
> > >   "The apex of each droplet's trajectory is visually identifiable as the highest point before descending."
> > > ],
> >
> >
> >
> > We found that directly depicting the physical phenomenon in the standards—rather than using abstract statements like “whether the physics is correct”—leads to substantially higher agreement among human annotators. When the criteria describe concrete, visually observable behaviors, it becomes much easier for both humans and MLLMs to determine whether the action is correct. This design reduces ambiguity, improves consistency across evaluators, and makes the evaluation process more robust in visually complex scenarios.
> >
> >
> >
> > ---
> >
> >
> > **Re: (3) Measuring whether the physical process is smooth, coherent, and temporally consistent**
> >
> > Our benchmark explicitly evaluates *both* the progression of the physical process and the final outcome. This is reflected throughout the `prompts-with-standard-and-index.json` file, where every case includes standards that capture temporal evolution, not just end states.
> >
> > For example, for prompt **014-1**, the prompt is:
> > **“A car drifts around a sharp turn.”**
> > The corresponding standards are:
> >
> > > "Basic_Standards": {
> > >   "Objects": [
> > >     "a car"
> > >   ],
> > >   "Event": "drifts around a sharp turn"
> > > },
> > > "Key_Standards": [
> > >   "The car follows a circular path.",
> > >   "The car's direction changes continuously during the turn."
> > > ],
> >
> > These standards clearly require assessing **the temporal trajectory** (e.g., continuous direction change, circular motion), not just the final frame. This demonstrates that our evaluation framework inherently measures smoothness, coherence, and temporal consistency across the entire physical process, rather than only checking the final result.
> >
> >
> >
> > And here is the output from CAP:
> >
> >
> > > ## What is happening in the video
> > > The video shows a blue sports car drifting aggressively along a sharp mountain road turn. The car slides sideways with heavy tire smoke, counter-steers through the curve, and maintains an arc-like trajectory while the camera captures multiple angles of the drift.
> > >
> > > ## Evaluation Against Standards
> > >
> > > ### 1. Basic Standards
> > >
> > > **Objects**
> > > - Required: a car
> > > - Observed: ✔ A blue sports car is clearly present.
> > >
> > > **Event**
> > > - Required: "drifts around a sharp turn"
> > > - Observed: ✔ The car is visibly drifting with significant smoke and oversteer through a tight mountain turn.
> > >
> > > **Result:** **PASS**
> > >
> > > ---
> > >
> > > ### 2. Key Standards
> > >
> > > **1. The car follows a circular path**
> > > - The vehicle moves along a curved mountain road and clearly traces a circular/arc-like trajectory while drifting.
> > > **PASS**
> > >
> > > **2. The car's direction changes continuously during the turn**
> > > - The car counter-steers and rotates smoothly throughout the drift, showing continuous change in orientation.
> > > **PASS**
> > >
> > > ---
> > >
> > > ## **Final Judgment**
> > > **PASS** — The generated video meets all Basic Standards and Key Standards.

---

### Official Review · Reviewer_oKee · 2025-10-31

**Soundness:** 2
**Presentation:** 3
**Contribution:** 3
**Rating:** 6
**Confidence:** 4

**Summary:**

Summary:
This paper addresses the critical limitation of existing text-to-video (T2V) generation models—their inability to adhere to physical laws despite producing photorealistic content—by proposingPhyWorldBench, a comprehensive benchmark for evaluating physical realism in T2V models.

Contributions:
（1）Development of a comprehensive physics benchmark: PhyWorldBench fills the gap of lacking holistic tools to evaluate physical realism in text-to-video models. Its hierarchical structure (three levels, 10 main categories, 50 subcategories) covers diverse physical scenarios—from basic motion and energy conservation to anti-physics scenarios—and includes 1,050 well-curated prompts with variations, enabling systematic testing of models’ physical reasoning capabilities.
（2）Creation of a zero-shot automatic evaluator: The proposed Context-Aware Prompt method resolves the limitations of traditional evaluation. By guiding large language models to explicitly assess AI-generated videos and use chain-of-thought reasoning, it achieves high accuracy in evaluating physical realism, providing a scalable and objective alternative to costly human evaluation.
（3）Extensive evaluation of state-of-the-art models: The paper conducts large-scale tests on 12 leading text-to-video models, generating 12,600 videos. This evaluation identifies key challenges models face—such as struggling with complex interactions and prioritizing cinematic aesthetics over physics—and quantifies performance gaps across different physics categories, offering clear directions for future model improvement.

**Strengths:**

Strengths:
（1）Originality
It proposes a three-tier (Fundamental, Composite, Anti-Physics) hierarchical structure with 10 main physics categories, going beyond prior benchmarks by using Anti-Physics scenarios to test true physical understanding. The CAP evaluator creatively combines LLMs with domain constraints, separating aesthetics from physics via two-step reasoning to fix traditional evaluators’ inaccuracies. It also offers three prompt variants to test model adaptability, filling gaps of static-prompt benchmarks.
（2）Quality
Its benchmark is built via a rigorous three-stage process (literature/expert category definition, LLM-human prompt generation, expert validation) to ensure 1,050 prompts are diverse and accurate. CAP aligns with human evaluation (ROC-AUC: 80.3 for SA, 75.1 for PC) and outperforms baseline LLMs. Experiments cover 12 models (5 proprietary, 7 open-source) with 12,600 videos and systematic analyses to avoid cherry-picking.
（3）Clarity
It follows a clear "problem-solution-validation" flow, with the introduction outlining existing benchmark limits and methodology linking components to specific issues. Technical details (CAP’s reasoning, "Yes/No" criteria) are explained in plain language without jargon. Consistent terminology and condensed key findings make it readable for physics and AI researchers.
（4）Significance
As a standardized tool, its open benchmark and CAP provide universal metrics for tracking text-to-video physical realism. Prompt design insights (e.g., Physics-Enhanced Prompts boost PC) guide real-world uses like scientific visualization. It shifts evaluation focus to physical correctness, identifies model weaknesses, and reduces educational misinformation risks.

**Weaknesses:**

Weaknesses:
（1）CAP Evaluator’s Aesthetic Bias
The Context-Aware Prompt (CAP) evaluator favors visually polished videos (e.g., smooth lighting, dynamic camera movement) over physically correct ones, which conflicts with the benchmark’s goal of prioritizing physical plausibility. However, the paper doesn’t quantify this bias or fix it. For example, a visually vivid but gravity-violating floating apple video might get a higher score than a plain yet physically correct one.
（2）Inadequate Niche Fundamental Physics Analysis
PhyWorldBench focuses on common physics (e.g., free fall) but ignores niche subcategories of fundamental physics, like pressure-dependent phase changes (water boiling at high altitude) or non-uniform heat transfer (a metal rod heating unevenly). It only reports broad category performance, hiding which specific principles models struggle with.
（3）No Long-Duration Video Validation
The paper doesn’t specify video duration or test how performance scales with length. Physical inconsistencies (e.g., trajectory drift) worsen in longer videos, which are needed for real uses (e.g., education). Without this data, the benchmark’s real-world value is limited.
（4）Lack of Specialized Benchmark Comparisons
The paper only compares to general physics benchmarks (e.g., VideoPhy) but not specialized ones like Morpheus (real-world experiment focus) or T2VPhysBench (first-principles physics). This hides PhyWorldBench’s unique value.

**Questions:**

Questions:
（1）Questions About CAP Evaluator’s Aesthetic Bias
You note the Context-Aware Prompt evaluator prefers visually polished videos but provide no quantitative data on this bias. How often does this preference lead to misclassifying physically incorrect yet visually polished videos as physically plausible? Does a visually vivid video with physical violations score higher than a plain but physically correct one? Have you tested if revising CAP prompts to explicitly ignore aesthetics reduces this bias? Clarifying these points will confirm if CAP’s objectivity is compromised and if adjustments can fix it.
（2）The comparison with the methods of predecessors is not sufficient.
A comparison between the dataset and some previous related datasets, such as VideoREPA[1], WISA[2], NewtonGen[3], etc.?
（3）Niche Fundamental Physics Performance
PhyWorldBench covers 10 main physics categories, but your experiments focus on common phenomena and lack breakdowns for niche subcategories like pressure-dependent phase changes or non-uniform heat transfer. Do you have data on model performance in these niche areas? If not, why were they excluded from detailed analysis? Understanding these gaps will help researchers target specific physical principles models struggle with. If the author's reply is strong and reasonable, I will consider increasing my score.

[1] VideoREPA: Learning Physics for Video Generation through Relational Alignment with Foundation Models
[2] Wisa: World simulator assistant for physics-aware text-to-video generation
[3] NEWTONGEN: PHYSICS-CONSISTENT AND CONTROL-LABLE TEXT-TO-VIDEO GENERATION VIA NEURAL NEWTONIAN DYNAMICS

---

> ### Author Response · Authors · 2025-11-26
> **Rebuttal 1/2**
>
> Thank you for the detailed and constructive review. We appreciate the reviewer’s thoughtful assessment of our contributions and the clear identification of areas for improvement. Below, we provide clarifications and additional results addressing each of the raised questions and concerns. The referred videos are updated in supplementary material.
>
> ---
>
>
> **Re: (1) The Context-Aware Prompt (CAP) evaluator favors visually polished videos**
>
> The concern primarily reflects limitations of the underlying LLM used for evaluation, rather than an inherent issue with CAP itself. As more capable video-understanding LLMs emerge, their ability to reason about temporal dynamics improves significantly, reducing reliance on surface-level aesthetic cues.
>
> In our updated experiments with GPT-5, we observe that the evaluator’s judgments align closely with human assessments and are not influenced by video polish. For instance:
>
> - **Video 001-1_open-sora-plan.mp4**:
>   GPT-5 correctly identifies the physics outcome despite the video’s good aesthetic score, consistent with the human rating.
>
> - **Video 001-3_hunyuan.mp4**:
>   Even though this video has higher aesthetic quality, GPT-5 accurately flags major physics violations—such as the rocket failing to fly in a straight line—leading to a lower physics score.
>
> We include the CAP outputs under GPT-5 to demonstrate that the evaluator tracks true physical correctness rather than visual aesthetics.
>
>
> First for video 001-1_open-sora-plan.mp4, the CAP output is:
>
> > ## What is happening in the video
> > The video shows a rocket lifting off from the ground with a bright exhaust plume. It ascends upward into the sky, maintaining a mostly straight vertical trajectory. The camera captures the rocket from a stable ground-up angle as it rises smoothly, with no sudden shifts in velocity visible across the frames.
> >
> > ## Evaluation Against Standards
> >
> > ### 1. Basic Standards
> >
> > **Objects**
> > - Required: a rocket
> > - Observed: ✔ A rocket is clearly present.
> >
> > **Event**
> > - Required: "rocket launches into the sky"
> > - Observed: ✔ The rocket visibly ignites, lifts off the ground, and ascends into the sky.
> >
> > **Result:** **PASS**
> >
> > ---
> >
> > ### 2. Key Standards
> >
> > **1. The rocket moves in a straight vertical line during its ascent**
> > - The rocket rises along a consistent vertical trajectory with no noticeable lateral drift.
> > **PASS**
> >
> > **2. The rocket's upward motion appears smooth, without abrupt changes in speed**
> > - The ascent shows a continuous, steady upward movement. No sudden accelerations or decelerations are visible.
> > **PASS**
> >
> > ---
> >
> > ## Final Judgment
> > **PASS** — The generated video meets all Basic Standards and Key Standards.
>
>
>
>
>
> For 001-3_hunyuan.mp4, the CAP output is:
>
>
>
> > ## What is happening in the video
> > The video shows a large multi-stage rocket launching from a ground pad at night. Bright exhaust flames illuminate the area as it lifts off. After ignition, the rocket rises but begins to tilt and follow a curved trajectory rather than ascending straight upward. A large plume of smoke expands across the launch site as the rocket climbs.
> >
> > ## Evaluation Against Standards
> >
> > ### 1. Basic Standards
> >
> > **Objects**
> > - Required: a rocket
> > - Observed: ✔ A rocket is clearly present.
> >
> > **Event**
> > - Required: "rocket launches into the sky"
> > - Observed: ✔ The rocket ignites and ascends into the sky.
> >
> > **Result:** **PASS**
> >
> > ---
> >
> > ### 2. Key Standards
> >
> > **1. The rocket moves in a straight vertical line during its ascent**
> > - The rocket **does NOT** move vertically.
> > - Its ascent follows a **curved, angled trajectory**, tilting noticeably shortly after lift-off.
> > **FAIL**
> >
> > **2. The rocket's upward motion appears smooth, without abrupt changes in speed**
> > - The ascent appears smooth and continuous, with stable acceleration.
> > **PASS**
> >
> > ---
> >
> > ## Final Judgment
> > **FAIL** — The video meets the Basic Standards but fails a Key Standard because the rocket does not ascend in a straight vertical line.

---

> ### Author Response · Authors · 2025-11-26
> **Rebuttal 2/2**
>
> **Re: (2) Niche Fundamental Physics Analysis**
>
> Our benchmark already covers a wide spectrum of physical phenomena compared with prior datasets. For example, we include niche scenarios such as *“a glass cutter scores a pane of glass”*, which go beyond what existing benchmarks attempt to capture. While it is inherently difficult to cover every specialized physics case in a finite test set, our benchmark’s scale, category diversity, and physical coverage are already substantially larger than those of previous datasets.
>
> ---
>
> **Re: (3) Lack of video duration specification and scaling analysis**
>
> Most video generation models today output videos with fixed, model-defined durations, which prevents us from standardizing or controlling video length across systems. Because of this limitation, we did not include experiments on how performance changes with longer videos. We agree this is valuable, especially since longer videos can expose more physical inconsistencies (e.g., drift in trajectories), and we plan to incorporate this when models support configurable durations.
>
> Below are the actual video lengths used in our evaluation (shown in a two-row table):
>
> | CogVideo | Gen-3 | Hunyuan | Kling | LTX-Video | Luma | Open-Sora-Plan | Open-Sora | Pika | Sora | Wanx | Step-Video-T2V |
> |---------|-------|---------|-------|-----------|------|----------------|-----------|------|------|------|----------------|
> | 5s | 5s | 5s | 5s | 5s | 5s | 5s | 2s | 5s | 5s | 5s | 8s |
>
> As shown, both open-source and proprietary models generally restrict users to default lengths (mostly 5 seconds) and do not allow customization. We have updated the paper to explicitly include these durations and to clarify why length-scaling experiments are not included at this stage.
>
>
> ---
>
>
> **Re: (4) Comparison with Morpheus, T2VPhysBench, VideoREPA, WISA, and NEWTONGEN**
>
> Thank you for the suggestion. We agree that Morpheus and T2VPhysBench are important and relevant specialized benchmarks, and we have updated the paper to discuss them explicitly. However, their goals differ from ours in fundamental ways:
>
> - **Morpheus** evaluates models using a small number of controlled laboratory-style experiments, relying on real-video conditioning and trajectory-invariant metrics. Its focus is orthogonal to ours, as PhyWorldBench is designed for prompt-based, broad-coverage evaluation of generative capability rather than controlled physical setups.
>
> - **T2VPhysBench** concentrates exclusively on named physical laws using 84 prompts. In contrast, **PhyWorldBench scales to 1,050 prompts** and targets realistic everyday physics, composite scenarios, and anti-physics cases that frequently arise in real-world generations. We have updated Table 1 in the main paper to include T2VPhysBench for completeness.
>
> Thus, PhyWorldBench fills a different gap: assessing whether text-to-video models exhibit *general physical commonsense* across diverse, realistic scenarios—not only canonical physics experiments or isolated laws.
>
> Regarding **VideoREPA**, **WISA**, and **NEWTONGEN**: these are video generation models rather than evaluation benchmarks. Since our paper focuses on benchmarking and evaluation, they were not discussed initially. We have now updated the paper to include them for clarity and completeness.

---

### Author Response · Authors · 2025-11-26
**General Response**

We sincerely thank all reviewers — **oKee, hYMe, FCTE, and aZkX** — for their thoughtful evaluations and constructive feedback. We are also encouraged by the broad set of strengths highlighted across the reviews, which validate the motivation, design, and contribution of **PhyWorldBench**.

Across reviewers, several core advantages of our work were consistently recognized:

- **Comprehensive and systematic coverage of physics phenomena** (#reviewer **oKee**, #reviewer **hYMe**, #reviewer **FCTE**, #reviewer **aZkX**)
  Reviewers noted that our three-tier structure (Fundamental, Composite, Anti-Physics), 10 main categories, 50 subcategories, and 1,050 validated prompts represent the most extensive and fine-grained organization among existing T2V physics benchmarks.

- **Strong, practical automated evaluator (CAP)** (#reviewer **oKee**, #reviewer **hYMe**, #reviewer **FCTE**)
  The CAP evaluator was highlighted for its context-aware reasoning, separation of aesthetics vs physics, and strong alignment with human judgments. Reviewers noted that this offers a scalable, reproducible, and cost-effective alternative to human studies.

- **Large-scale and rigorous experimental study** (#reviewer **oKee**, #reviewer **hYMe**, #reviewer **FCTE**, **aZkX**)
  Multiple reviewers highlighted the comprehensiveness of evaluating 12 models, generating 12,600 videos, and conducting large-scale human annotation, as well as the diagnostic insights (e.g., difficulty in complex interactions, model reversion during anti-physics tasks).

- **Novel inclusion and utility of the Anti-Physics category** (#reviewer **oKee**, #reviewer **hYMe**)
  They emphasized that Anti-Physics tasks provide a unique way to test true physical understanding instead of memorized realism, addressing limitations in prior benchmarks.

- **Clarity and readability of the paper** (#reviewer **oKee**, #reviewer **FCTE**)
  Reviewers praised the intuitive structure (“problem → solution → validation”), clear explanation of CAP, minimal jargon, and consistently presented key insights.


We are grateful for these positive assessments and for the reviewers’ acknowledgement that PhyWorldBench offers a meaningful and timely contribution to the community. In the individual sections, we address each reviewer’s concerns in detail. We have also updated the paper to include the feedback from reviewers.

---

### Meta-Review · Area_Chair_FShU · 2025-12-13

**Summary:**

The paper introduces the PhyWorldBench, a benchmark for evaluating the Text-to-Video generative models for the ability to represent physical phenomena. The paper addresses the challenge that Text-to-Video models are notoriously struggling with accurate physics.

Strengths that warranty acceptance:
* The paper proposes a comprehensive evaluation framework for the fundamental issue of generative models, which is the ability to model physics.
* An extensive benchmark covering three aspects: Fundamental, Composite, and Anti-Physics, split into 10 main categories. The anti-physics category is introduced to access if the model can follow introductions that intentionally violate physics
* The paper introduces a novel way to automatically evaluate physical correctness of generated videos based on Large Language Models and show that it is correlated with human judgement.
* The paper evaluated a wide range of proprietary and open-source models.

**Reviewer Concerns:**

The authors have defended the following claims:
* Aesthetic Bias in Evaluator: authors show that using more advanced LLMs like GPT-5

* No Long-Duration Video Validation. Authors make a good point that most video models are limited to short duration without an ability to extend.

* Comparison to other physics benchmarks. The  authors updated the paper to include the comparisons and argued that the new benchmark is more comprehensive than the previous works.

* The authors provided the code for the standalone pipeline.

* Novelty of Prompt Granularity: the authors clarify that mentioning physical phenomena itself improves scores.

Remaining unaddressed questions:

* Lack of evaluation for longer videos and covering niche physics use cases. In the view of AC, these will be great directions for future work but do not affect the scores of the current paper.

**Reviewer Scores:**

Contribution of reviewer-author discussion to meta-review: In the assessment of AC, the paper is a novel and comprehensive contribution, tackling a key challenge in current T2V generative models. This strong foundation supported the initial acceptance decision. The authors’ rebuttal then successfully resolved the remaining open questions, reinforcing this positive recommendation.

How the reviewer would have changed their score if they had been able to participate fully in the discussion:
* Reviewer aZkX. Rating: 4 -> changing to 6
* Reviewer FCTE. Rating: 6 -> changing to 7
* Reviewer hYMe. Rating: 6 -> changing to 7
* Reviewer oKee. Rating: 6 ->  changing to 6

=======================================

**Note to authors:** please fix the following citation to the paper:

*Jiaqi Liao, Xinyu Tan, Wenqi Shao, Kaipeng Zhang, Yu Cheng, Dianqi Li, and Ping Luo. Devil:
A comprehensive benchmark for dynamics evaluation in video generation. arXiv preprint
arXiv:2407.01094, 2024*

The correct paper name and author list are the following:

*Mingxiang Liao, Hannan Lu, Xinyu Zhang, Fang Wan, Tianyu Wang, Yuzhong Zhao, Wangmeng Zuo, Qixiang Ye, Jingdong Wang.  Evaluation of Text-to-Video Generation Models: A Dynamics Perspective. arXiv preprint
arXiv:2407.01094, 2024*
https://arxiv.org/abs/2407.01094

---

### Decision · Program_Chairs · 2026-01-26

Accept (Oral)